# Compositional data analysis enables statistical rigor in comparative glycomics

Alexander R. Bennett [1], Jon Lundstrøm [2,3], Sayantani Chatterjee[4], Morten Thaysen-Andersen [4,5] & Daniel Bojar [2,3] ✉

Comparative glycomics data are compositional data, where measured glycans are parts of a whole, indicated by relative abundances. Applying traditional statistical analyses to these data often results in misleading conclusions, such as spurious "decreases" of glycans when other structures increase in abundance, or high false-positive rates for differential abundance. Our work introduces a compositional data analysis framework, tailored to comparative glycomics, to account for these data dependencies. We employ center log-ratio and additive log-ratio transformations, augmented with a scale uncertainty/information model, to introduce a statistically robust and sensitive data analysis pipeline. Applied to comparative glycomics datasets, including known glycan concentrations in defined mixtures, this approach controls false-positive rates and results in reproducible biological findings. Additionally, we present specialized analysis modalities: alpha- and beta-diversity analyze glycan distributions within and between samples, while cross-class glycan correlations shed light on previously undetected interdependencies. These approaches reveal insights into glycome variations that are critical to understanding roles of glycans in health and disease.

Glycomics, the comprehensive study of glycan structures within a biological sample[1,2], yields fundamentally compositional data. While "composition" in the context of a glycan may refer to its underlying monosaccharides (e.g., Hex3HexNAc4), within statistics it refers to a fixed whole that contains several components with non-negative quantities (e.g., relative abundances in the context of glycomics). Within this work, we use the statistical meaning of "composition" unless specified otherwise. Using such compositional data, comparative glycomics, which aims to quantitatively compare the measured glycome between two or more conditions, has emerged as an increasingly popular approach to unveil glycan abundances differences across disease conditions[3–5].

These data, representing relative abundances of glycans, reside on the Aitchison simplex−a geometric representation where the whole is the sum of its parts. Compositional data are ubiquitous in systems biology[6–9] and are found whenever (implicit) proportions or probabilities are the subject of analysis[10]. This extends to sequencing-based areas, wherein the total number of reads is immaterial and only relative proportions (between genes or between samples) are relevant. In the context of glycomics data, relative abundances usually indicate the proportion of integrated ion intensity of one glycan species in relation to the total glycan-derived ion intensity measured by mass spectrometry. Changes in relative abundances of glycans are often cited as potential disease biomarkers[11,12]. Yet not accounting for the compositional nature of glycomics data is a major contributor to divergent results of differential expression methods[13] and, in general, results in very high false-positive rates[14,15]. Traditional comparative glycomics analyses typically ignore these compositional characteristics, leading to spurious interpretations and false-positive results, such as perceived decreases or increases in glycan quantities

[1]Department of Medical Biochemistry, Institute of Biomedicine, University of Gothenburg, Gothenburg, Sweden. [2]Department of Chemistry and Molecular Biology, University of Gothenburg, Gothenburg, Sweden. [3]Wallenberg Centre for Molecular and Translational Medicine, University of Gothenburg, Gothenburg, Sweden. [4]School of Natural Sciences, Faculty of Science and Engineering, Macquarie University, Sydney, NSW, Australia. [5]Institute for Glycocore Research (iGCORE), Nagoya University, Nagoya, Japan. ✉e-mail: daniel.bojar@gu.se

that are artifacts of relative abundance changes rather than absolute ones.

To make this point clear, the addition of an exogenous glycan standard in high concentration to one sample would lead to the perceived "downregulation" of all other present glycans in that sample, as their relative abundances decrease. This misinterpretation stems from an oversight of the simplex's constraints, where an increase in one part necessitates a decrease in another. It is a mathematical imposition, not a biological phenomenon, and necessitates a tailored analytical approach to focus the findings on biological effects instead. Compositional data analysis (CoDA)[16] provides such a framework, respecting the relative scale of the data and avoiding the misapprehensions that traditional methods incur.

Our work presents a CoDA-based framework specifically designed for glycomics (N-linked, O-linked, glycosphingolipids (GSLs), milk oligosaccharides, etc.), which we show to be also applicable to glycoproteomics, the system-wide analysis of intact glycopeptides[17]. Central to our approach are the center log-ratio (CLR) and additive log-ratio (ALR) transformations. The former normalizes glycan abundances to the geometric mean of a sample, facilitating comparisons across conditions as the relations between the individual abundances are accounted for. The latter normalizes abundances to a rigorously chosen reference glycan that best re-captures the geometry achieved by CLR transformation. We then further refine these transformations by integrating both uninformed (i.e., scale uncertainty) and informed-scale models to account for changes in absolute scale between conditions[18,19] (i.e., greater or lower number of glycan molecules overall, in one condition). This, together with other improvements, markedly enhances the sensitivity and robustness of glycomics data interpretation. We present our pipeline(s), integrated into the open-source glycowork Python package[20], as the state-of-the-art in comparative glycomics data analysis.

When applied to a range of glycomics datasets, we show that differential expression analysis using this enhanced CLR/ALR transformation can reveal biological insights. We then further improved analysis modalities by developing CoDA-based glycome similarity analyses using CoDA-appropriate alpha- and beta-diversity metrics, such as Aitchison distance[6]. These have revealed variations within and between biological samples, enriching our understanding of glycan roles in physiological processes. Lastly, we introduce a CoDA-based method for assessing cross-class glycan correlations, similar to Sparse Correlations for Compositional Data (SparCC)[21], originally developed for microbiome data analysis. This has provided a window into complex, previously concealed, glycan interdependencies, as well as glycan motif–transcript associations.

The introduction of these methodologies represents a significant advancement in comparative glycomics data analysis and is in line with the need for advanced data science approaches in the glycosciences[22]. For maximum uptake and impact, all methods are integrated into the glycowork package (version 1.3+), presenting a full analysis suite for glycomics data that adheres to the proper analysis of compositional data. By providing a more accurate interpretation of glycan systems biology data, our approach facilitates the discovery of robust biological insights with implications for understanding the roles of glycans in health and disease. This analytical paradigm promises to redefine comparative glycomics data analysis, offering a rigorous foundation for future explorations into the glycome.

## Results

### Analyzing comparative glycomics data as non-compositional data is fundamentally flawed
The current gold standard in analyzing glycomics data is to express the individual glycans as relative abundances (e.g., as percent of the total ion intensity) and then perform individual statistical tests for each glycan between conditions. As will be shown in the following, this is a fundamentally flawed approach that is both lacking sensitivity as well as incurring an intolerably large fraction of false positives. Two interconnected reasons can be brought forth to explain this. (i) The interdependent nature of relative abundances means that an increase in glycan A demands a decrease in all other glycans—even if these other sequences exhibit a constant number of molecules across conditions (Fig. 1a, b, Supplementary Data 1), causing spurious findings and obscuring real findings. (ii) Additionally, as will be shown, the total number of molecules is rarely invariant across conditions, further distorting the obtained relative abundances and their differences across conditions.

These limitations are, in part, known to practitioners, yet are commonly ignored in practice. Occasionally, researchers will analyze ratios between glycans across conditions, perhaps out of the correct—but tacit—intuition that this mitigates the compositional nature of the data. We maintain here that this compositional nature, if not addressed correctly, is a major problem in the field and will become worse as sample sizes increase, because of unacknowledged bias[14,18], leading to incredible false-positive rates of >30% even at rather modest sample sizes (Fig. 1c). Statistically, the interdependent nature of relative abundances can be resolved with transformations such as CLR or ALR[6], transforming the data from the Aitchison simplex to real space. The further inclusion of a simple scale uncertainty model (i.e., acknowledging uncertainty as to potential differences in the total number of molecules between conditions) then largely resolves the possibility of scale differences mentioned above[18,19]. Thus, our workflow for differential glycomics expression analysis, incorporating these principles, is robust to false-positive rates (Fig. 1c), while still maintaining excellent sensitivity (Fig. 1d), presenting a solid state-of-the-art foundation for the burgeoning exploration of glycomics and the role of glycans in diseases.

For a standard, two-group, differential glycan profile analysis, our workflow (Fig. 1e) will automatically infer whether to use ALR or CLR for data transformation (Fig. 1f; see "Methods"; mainly dependent on the presence of a suitable reference component for ALR). On the one hand, this is coupled with outlier treatment, machine learning-based imputation, variance-based filtering, and multiple testing correction[5]. On the other hand, this also supports analyses on the sequence, motif, and motif set level[5]. While we expect further advances to tailor these methods more to other types of data, such as glycoproteomics, glycolipidomics, or lectin microarray analysis[23], we note that this workflow is applicable to glycoproteomics data already, without any required changes (Supplementary Fig. 1).

### A compositional data analysis workflow for comparative glycomics unveils biological insights
Another corollary of compositional data is that real-space distance metrics (e.g., Euclidean distance) are not valid in the Aitchison simplex[6]. Thus, we reasoned that clustering—usually based on distance matrices—can be improved by CLR/ALR transformation. Applying this to a bacteremia N-glycomics dataset[3], we indeed find that Aitchison distance (i.e., Euclidean distance after ALR transformation) results in an improved clustering (Supplementary Fig. 2) that better separates patient and donor classes than clustering on the distances between log-transformed glycan abundances (adjusted Rand index: 0.79 vs 0.74; normalized mutual information: 0.76 vs 0.70). We are further excited to see even finer clustering below disease classes using Aitchison distance, such as a sub-clustering of the healthy volunteers in this sample by sex, which is known to affect glycosylation profiles[24].

We then used this insight to reanalyze another dataset, of human B-cell O-glycans from acute lymphoblastic leukemia patients and healthy bone marrow donors[25], illustrating the effectiveness of compositional data handling in making comparison between samples (Fig. 2a). The aim of this study was to identify glycoprotein markers of

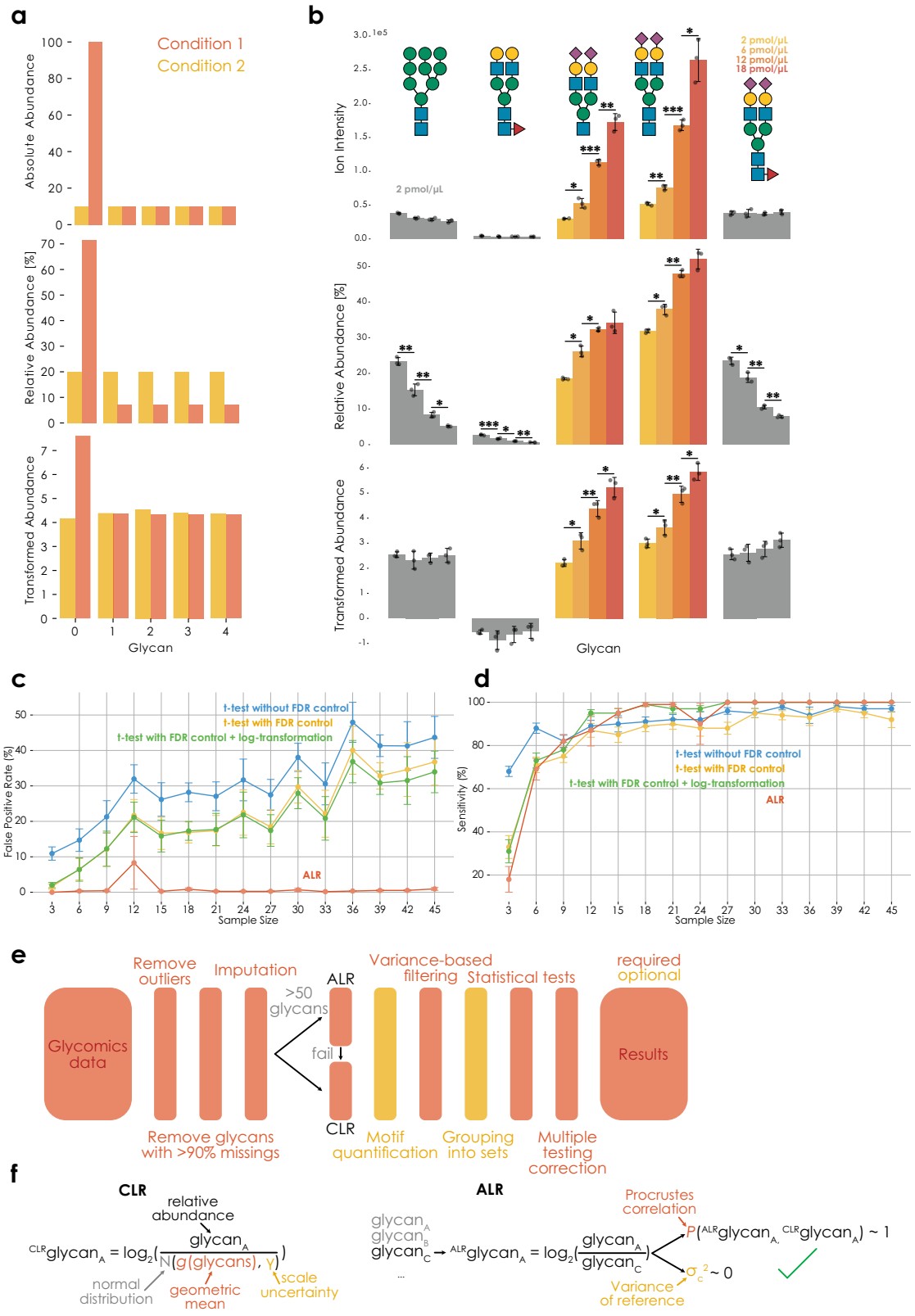

diagnosis and prognosis of mixed-lineage leukemia gene rearranged B-cell precursor acute lymphoblastic leukemia. Healthy bone marrow samples (RB7) were compared to MLL samples, and comparative gly-comics was employed to propose biomarker candidates. Upon rea-nalysis of these data with CLR transformation, we find much more effective clustering between glycans and samples, compared with the same clustering approach using relative abundances (Dunn index

0.828 vs 8.647, which is also higher than the Dunn index of 7.928 that is achieved by log transformation alone).

Additionally, we have analyzed a dataset of GSLs derived from ocular tissues of seven human donors[26]. This exploratory study aimed to characterize and compare the gangliosides of several ocular and peripheral tissues. Here, we find improved statistical power to detect whether tissues differed in mono-, di-, and tetra-sialylated GSLs,

**Fig. 1 | Glycomics data are compositional. a** Relative to absolute abundances (top), relative abundances (middle) distort relationships between glycans, which can be rescued via compositional data analysis (bottom). Transformed abundances indicate CLR-transformed data with an informed-scale model. **b** Ion intensities (absolute values), relative abundances (%), and ALR-transformed abundances of glycan standards (Supplementary Data 1). Control standards (gray) were added as 2 pmol/μL in all samples, whereas standard 3/4 were added in increasing concentrations (colored). Data are shown as bar graphs with standard deviation (centered at sample mean), with data points ($n = 3$ independent mixtures) overlayed as a scatterplot. Significance was established via two-tailed Welch's $t$-tests. All comparisons without a star did not reach the significance threshold (<0.05). **c** False-positive rate (FDR) for identifying spurious differences between conditions increases with sample size when ignoring the compositional nature of glycomics data. Glycomics data were simulated via Dirichlet distributions with defined effects (see "Methods"). For each sample size, 10 independent simulations were performed ($n = 10$) and results are shown as means with standard deviation as error bars. The

Benjamini−Hochberg procedure was used for FDR control, except for ALR (our herein presented workflow, which used the two-stage Benjamini−Hochberg procedure). **d** Compositional data analysis maintains excellent sensitivity for identifying differences between conditions. Simulations as in (**c**), with the difference of tracking sensitivity instead of type I error. **e** Overview of CoDA-improved workflow for differential glycomics abundance analysis. While the schema most closely resembles the differential expression workflow (*get_differential_expression*), indicated steps were largely preserved in all analyses presented here and in glycowork. ALR was considered to have failed if the Procrustes correlation of the best reference glycan was below 0.9 or if its $\log_2$-transformed variance was above 0.1. **f** Implementing CLR- and ALR-transformations for glycomics data. ALR depicts the successful choice of a reference glycan, which fulfills the desired criteria. Scale uncertainty can further be introduced into ALR by subtracting $\log_2(N(0, \gamma))$ from the reference. Both depictions describe the usage with an uninformed scale model (i.e., scale uncertainty). *$p < 0.05$, **$p < 0.01$, ***$p < 0.001$. Source data are provided as a Source Data file.

whereas trisialyl species did no longer show differential tissue expression when using CoDA-based analyses (Fig. 2b). This also showed that our workflow is applicable to glycolipid data, extending beyond glycans liberated from protein carriers.

We then extended these re-analyses to a large set of comparative glycomics studies (Table 1), where we overall show that CoDA-based analyses often differed from analyzing relative abundances, indicating the importance of appropriately analyzing glycomics data for valid conclusions. Importantly, even for previous studies without formal statistical analyses, we could confirm and/or revise qualitative conclusions that the respective authors drew from their glycomics data.

### Informed-scale models for glycomics shed light on absolute changes in glycans

The scale uncertainty model presented above is an effective means to prevent false positives and can be universally applied by choosing one simple variance parameter $\gamma$. As pointed out by others[19], this means that this $\gamma$ parameter, when varied, can even serve as an indicator of effect robustness (i.e., an effect is deemed more robust if it is preserved under a higher $\gamma$). We probed this phenomenon with an analysis of a skin cancer $O$-glycomics dataset[27]. For each structure, we report the $\gamma$-cutoff value at which no significant difference can be detected (Supplementary Fig. 3), providing robustness indicators for each difference and showing that the dysregulation of the disialyl T antigen presented the most robust finding in this case.

The default values of $\gamma$, while introducing some variance, still routinely result in reproducible analyses, even if the random seed is not fixed (Supplementary Fig. 4). However, as practically useful as scale uncertainty is, higher values of $\gamma$ will also inevitably lower the sensitivity of analyses (Supplementary Fig. 3). We thus investigated whether we could also achieve an informed-scale model[14], injecting information into our analyses about the actual difference in terms of numbers of molecules between conditions, instead of merely introducing uncertainty about this difference. Many sources for this informed-scale model could be envisioned, yet−since this information is often available to researchers−we started by using the sum of integrated ion intensities from the original glycomics raw data as a measure of the total glycan signal within a sample.

Of course, quantifying glycans via MS1 peak integration is inevitably confounded to some degree by differences in ionization propensity of different glycans and glycopeptides[28], which is why it is crucial to include an error term in this scale estimation to account for such effects. It is important to note, however, that the differential ionization propensity of glycans (e.g., sialylated glycans ionizing more efficiently in negative ion mode) does not affect comparison of the same glycan across samples and thus is not necessarily a confounder in this context.

It is crucial to emphasize that this is only a feasible route if glycans from the same amount of (glyco)protein starting material have been analyzed for each sample, as ion intensities only then can serve as an indicator of scale. Following previous work[14], we then transform the relative abundances via the ratio of scales between groups, taking into account experimental error in determining those ion intensities in the first place, with the $\gamma$ parameter used before (Fig. 3a).

When applying this approach to comparative glycomics data, we routinely identified scale differences between conditions (e.g., Supplementary Fig. 5), showcasing how crucial the inclusion of an informed-scale model is for accurate biological conclusions. With the example of a longitudinal dataset of $N$- and $O$-glycomes during monocyte-to-macrophage differentiation[29], we decided to showcase this scale difference for time series analysis (full results in Supplementary Data 2−9). This study tracked changes in the $N$- and $O$-linked glycome of monocytes during differentiation into macrophages, identifying glycans that were differentially or stably expressed during the differentiation process. Reassuringly, some effects, such as an increase in Neu5Acα2-3Gal substructures in $O$-glycans during differentiation, were generally maintained with an informed-scale model (Fig. 3b, c). Yet, as in this case, the total glycan signal increases over differentiation, and some results changed, compared to CLR/ALR-transformed data, with a scale uncertainty model. Examples here include changing results, such as a reported decrease of core fucosylation in $N$-glycans[5,29] that can no longer be observed when scale is considered (Fig. 3d, e), as the absolute trend now rather pointed to a, marginally significant, increase in core fucosylation. On the other side, additional results may emerge, such as an absolute increase of Neu5Acα2-6Gal termini in $N$-glycans during differentiation, yet only when scale is considered (Fig. 3f, g). It is important to note that, despite the increase in scale over time, some decreases in abundance were still maintained under this informed-scale model (Supplementary Fig. 6), potentially indicating an actual decrease in the number of specific molecules over the differentiation course.

As performed by some practitioners in the field, another possible avenue could have been to perform analyses directly on the raw ion intensities instead of relative abundances, possibly after log transformation. However, we find that the variance of ion intensities typically outweighs the additional information that they contain (Supplementary Fig. 7) and hence propose that relative abundances, combined with an informed-scale model that is derived from ion intensities, are a superior approach to analyzing comparative glycomics data.

### Analyzing glycome similarity and cross-class regulation as fresh views on differential glycome profiling

Many biological questions cannot be answered by a differential expression approach, which is why we set out to develop methods that

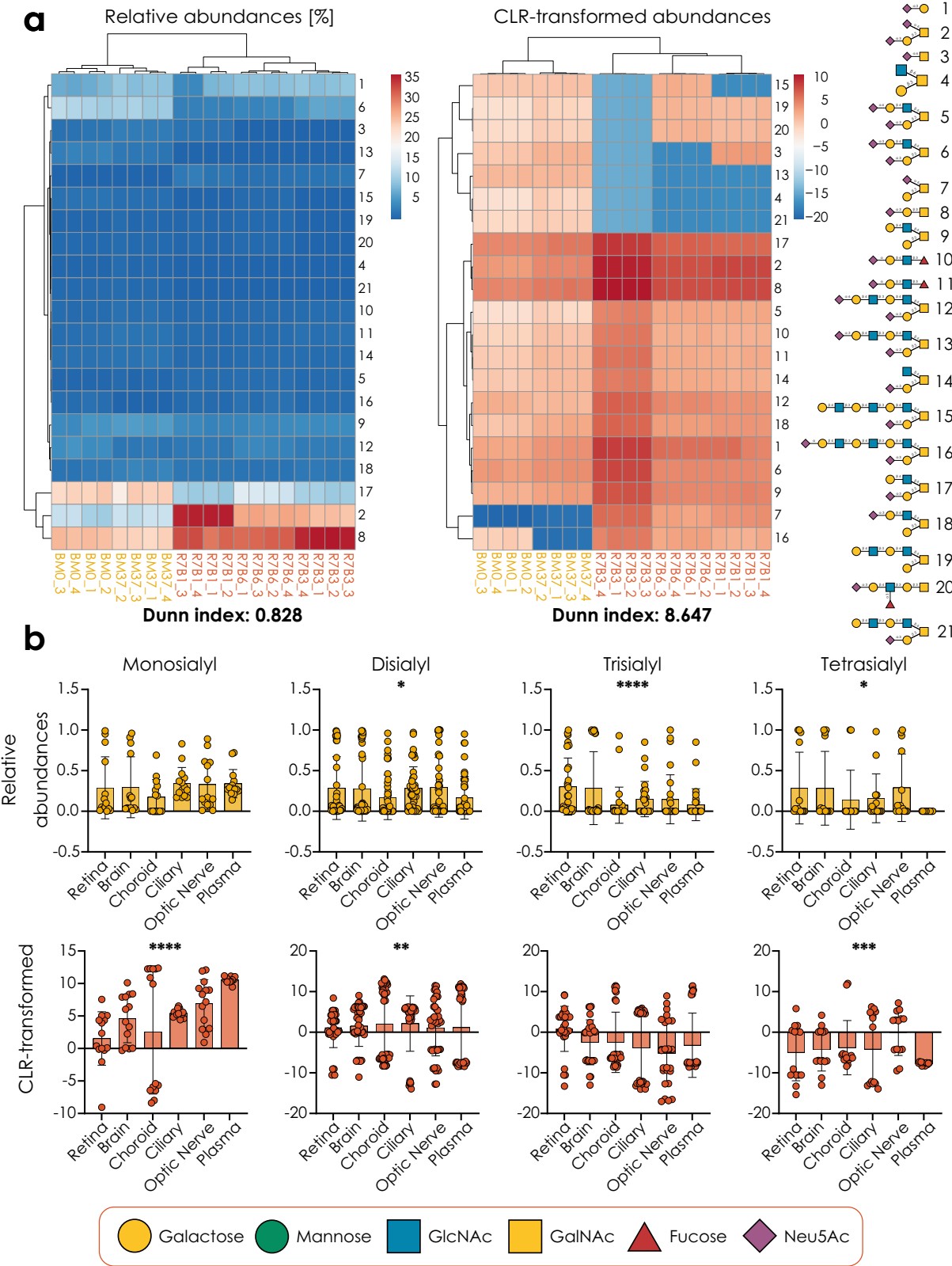

**Fig. 2 | Improved findings with CLR/ALR-transformed glycomics data.**
**a** Heatmaps of *O*-glycomics data of B-cells from acute lymphoblastic leukemia patients (cases labeled R, in red) and healthy bone marrow donors (cases labeled BM, in yellow)[25], with and without CLR transformation, illustrating differences in the effectiveness of heatmap clustering. Clustering was visualized using PHeatmap in R (version 4.3). **b** The abundance of mono-, di-, tri-, and tetra-sialylated glyco-sphingolipid species in ocular-related tissues from seven human donors[26] (*n* = 7),

presented as relative abundances and CLR-transformed relative abundances, with standard deviations depicted via error bars. Significance was determined via a Friedman test (two-tailed). All glycans in this work are presented via the Symbol Nomenclature for Glycans (SNFG) and are depicted with GlycoDraw[57]. *$p < 0.05$, **$p < 0.01$, ***$p < 0.001$, ****$p < 0.0001$. Source data are provided as a Source Data file.

**Table 1 | Findings upon compositional data-based analysis (CoDA) of published glycomics datasets**

| ID | Original findings | CoDA findings |
|---|---|---|
| PMID35112714[30] | Transfection of HEK293 cells with HIV-1 Gag and mock plasmids results in differential glycosylation signals. (No statistics reported regarding glycome composition) | Differential expression detected between only the non-transfected and Gag-transfected cells within the N-glycome ($p < 0.012$). Non-transfected cells show differential expression compared to both mock and gag-transfected cells within the O-glycome ($p < 0.0015$). |
| PMID38343116[37] | Sequential sample preparation results in greater sensitivity of analysis, demonstrated by greater numbers of glycans identified in comparison with parallel sample preparation, evidenced by total number of glycan species and peak areas. | Higher alpha diversities among sequentially prepared samples ($p < 0.004$) support original conclusions about sensitivity. |
| PMID26085185[53] | Tumorigenic tissues possess greater α2,6- and reduced α2,3- sialylation of N-glycans compared to non-tumorigenic controls ($p < 0.05$). | No significant differences among N-glycan motifs detected, including Neu5Ac-containing motifs. |
| PMID35832079[54] | One structure and 8 motifs were found to be differentially expressed between cancerous and healthy tissues. | 14 structures and 17 motifs were found to be differentially expressed between cancerous and healthy tissues. |
| PMID19152289[55] | Core 3 disialyl-Le$^x$ may be a biomarker of malignancy in colon cancer. | No significant difference is observed between the expression of core 3 disialyl-Le$^x$ structures in tumorigenic and health samples. |
| PMID34646384[25] | Non-significant differences observed in N-glycan sialylation and fucosylation in leukemia cells, along with increased core 2 O-glycans. | Significant differences observed in three sialylated/ fucosylated N-glycan structures, but no significant differences among N-glycan motifs. Significant differences in fucosylation, sialylation, and core 2 O-glycans. |
| PMID26075384[32] | Lewis structures can be used to differentiate mucinous and serous epithelial ovarian tumors. | Many motifs, including Lewis X, are significantly differentially expressed between the two tumor types. |
| PMID36952551[56] | N- and O-glycan sialylation decreases over time, as samples are stored at 22 °C. | No significant changes observed in structures or sialylated motifs. |
| PMID27997589[26] | Qualitative judgements on tissue specificities of ganglioside glycan chains, no statistical analysis is reported. | Significant differences in the expression of mono-, di-, and tetra-sialylated GSLs between tissues. |

Original findings from the respective publications were summarized and juxtaposed with results from corresponding CoDA using CLR/ALR transformation and a scale uncertainty model.

incorporate the CoDA principles laid out above. Example applications for this include questions related to overall glycome changes beyond individual sequences/motifs, biomarker discovery, or regulation across data modalities. To demonstrate the capabilities of these data analysis modalities, we then set out to reanalyze suitable glycomics datasets to gain robust insights. A study[30] investigating the impact of HIV *gag*-transfection on the *N*- and *O*-glycomes of cells additionally used a mock-transfected group as a procedural control. While differential expression analysis revealed significant differences for one *N*-glycan motif (antennary fucosylation; Supplementary Data 10) between the non-transfected and mock-transfected control groups, our alpha biodiversity indices reported no significant differences (Fig. 4a, b), suggesting that the differentially expressed motifs are of relatively low abundance. This corroborates the conclusions of the published study[30], which is that the mock-transfected cells were largely representative of the non-transfected cells, and the transfection process itself had only a minor influence on the glycome.

Biomarker discovery rests on identifying molecular features that reliably distinguish, via their abundance, between two or more states. In the context of a glycan, a feature can refer to a full glycan molecule or any of the substructures found in such molecules (e.g., fucosylation, Lewis X, level of branching in *N*-glycans, etc.) Glycan features have been emerging as very sensitive biomarker candidates[11], yet many glycan alterations are due to systemic changes (e.g., inflammation[31]) and thus potentially not very specific to a particular condition. To enable biomarker discovery within our CoDA framework, benefiting from an informed-scale model, we developed a receiver operating characteristic–area under the curve (ROC-AUC) pipeline to first identify the best distinguishing substructure between healthy and infected individuals from a recent dataset[3], where serum *N*-glycomics was performed to identify potential strain-specific biomarkers during bacteremia. Our analysis highlighted terminal Neu5Acα2-6Gal epitopes with an AUC of 0.91 (Fig. 4c). Then, addressing the issue of specificity of glycan-derived biomarkers, we engaged in a One-vs-Rest approach of the same function, uncovering the best glycan substructure to identify each specific bacterial infection, resulting in excellent biomarker candidates for each of the assayed bacterial

species (AUC 0.91–1.00; Fig. 4c) and extending the findings of the original study[3].

We then set out to show that glycomics beta-biodiversity—changes in glycan distribution measured by Aitchison distance[6]—is a potent means for clustering subjects from an ovarian cancer dataset[32]. In this dataset, glycoprotein secretions from serous and mucinous tumors of different grades were analyzed, to determine biomarker candidates of disease class and severity. A principal component analysis of the beta-diversity distance matrix indeed clearly showed the simultaneous clustering of mucinous and serous tumors as well as their respective high- and low-grade subgroups (Fig. 4d), which was quantifiably superior to clustering distances from un-transformed abundances (Dunn index and silhouette width in Fig. 4d). Notably, the first principal component here stratified mucinous and serous tumors, while the second principal component effectively separates high- and low-grade tumors, suggesting that the same changes in the glycome differentiate high- and low-grade tumors in distinct tissues.

Lastly, we wanted to address the, somewhat artificial, separation of the glycome into different classes, largely driven by methodological constraints. Thus, we sought to establish a cross-class analysis modality by developing a CoDA-based correlation analysis across glycan classes, conceptually similar to SparCC[21]. For this, we again investigated correlations between structural motifs associated with *N*- or *O*-glycans during macrophage differentiation[29] (Fig. 4e). *N*-linked sialylation, LacNAc extension, and biantennary complex structures showed positive correlation with sialylated and Fucα1-2Gal-containing *O*-glycans, and negative correlation with core 2-related structural motifs. The latter, in contrast, were found to positively correlate with high-mannose structures as well as *N*-glycans modified with core fucosylation or bisecting GlcNAc. Overall, while sialylation was positively correlated across classes, fucosylation showed no significant correlation between *N*- and *O*-glycans. Interestingly, we even identified a negative correlation between the presence of Lewis X motifs in *N*- and *O*-glycans, respectively.

Since we designed this workflow to be maximally flexible, we could use the same method to analyze cross-correlations between glycomics and other data types. We used this to investigate

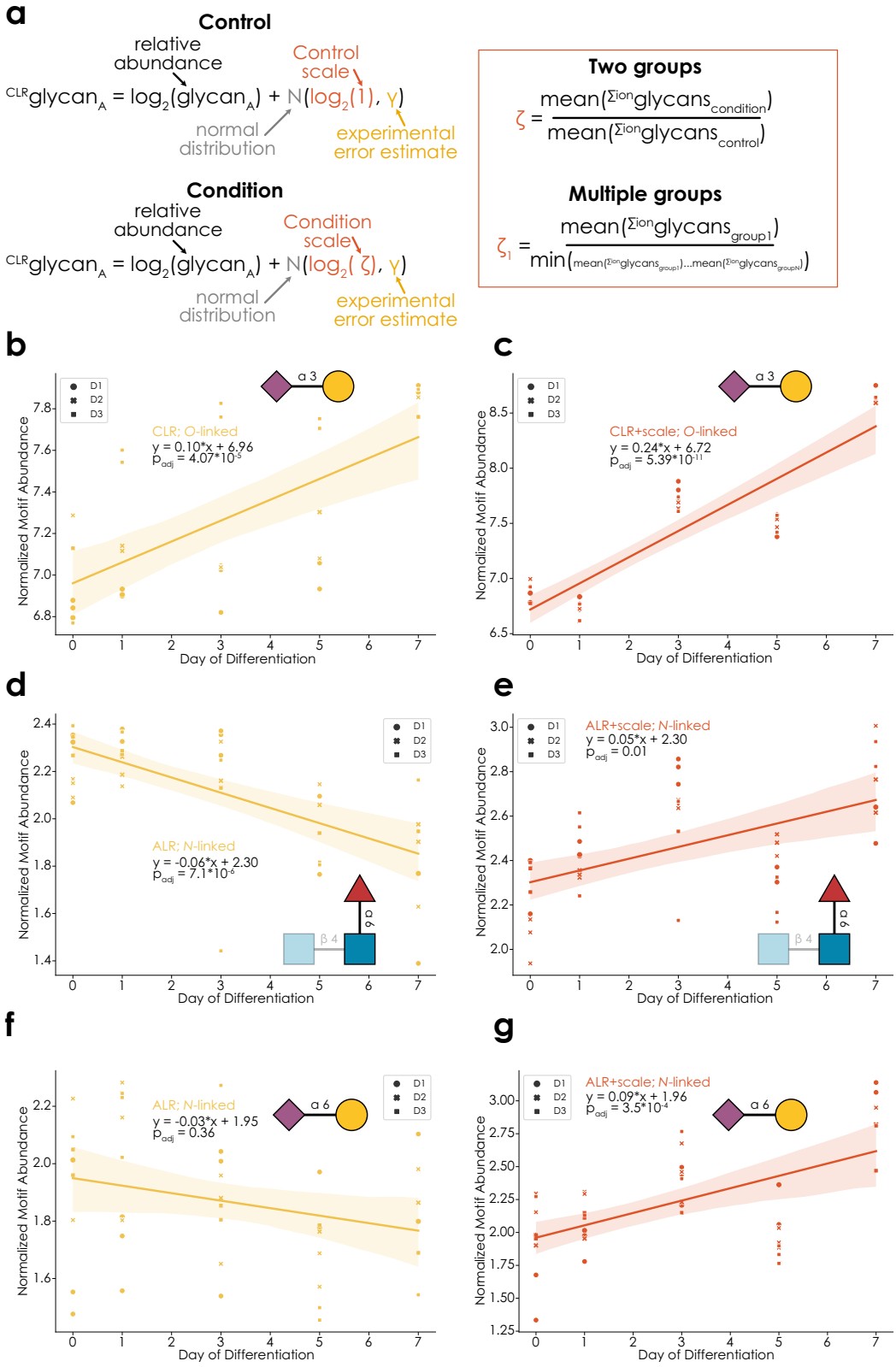

glycan motif–transcript relationships in the abovementioned B-cell dataset (Supplementary Fig. 8), still in the herein developed CoDA framework, now applied to both glycomics and transcriptomics data. This revealed significant associations of glycogenes and their cognate motif (e.g., ST6GALNAC1 and Neu5Acα2-6GalNAc in O-gly-cans, FUT4 and Fucα1-3GlcNAc in N-glycans, or GCNT4 and the core 2 structure in O-glycans) but also interesting negative associations

(e.g., the N-glycan branching enzyme MGAT4A and fucose-containing motifs) potentially hinting at motifs interfering with the action of an enzyme, such as branching inhibiting antennary fucosylation (or vice versa). Overall, our SparCC-inspired analysis of cross-class corre-lations can yield insights into possible co-regulatory mechanisms that would otherwise remain hidden if each dataset is considered independently.

**Fig. 3 | Informed-scale model with ion intensities. a** Applying an informed-scale model to glycomics data analysis. Terms within the equations are explained where necessary. Calculations of the scale factor $\zeta$ are provided for two as well as multiple groups and use the integrated ion intensities. The meaning of $\gamma$ in this case is transformed to an estimate of the experimental error in determining the absolute scales. In ALR, an informed-scale model is achieved by subtracting $\log_2(N(\zeta, \gamma))$ from the reference. **b, c** Monitoring Neu5Acα2-3Gal abundance in *O*-glycans during macrophage differentiation[29]. CLR-transformed relative abundances with a scale uncertainty (**b**) or informed-scale model (**c**), were used with the *get_time_series* (glycowork, version 1.3) function to fit an ordinary least squares model, where a two-tailed *F*-test determined the statistical significance of a linear regression. Donor identity (D1, D2, D3) for each data point is indicated by a unique shape. Monitoring core fucose (**d, e**) and Neu5Acα2-6Gal (**f, g**) abundance in *N*-glycans during macrophage differentiation. ALR-transformed relative abundances with a scale uncertainty (**d, f**) or with an informed-scale model (**e, g**) were used with the *get_time_series* (glycowork, version 1.3) function to fit an ordinary least squares model, where a two-tailed *F*-test determined the statistical significance of a linear regression. **b**–**g** Variance between samples by a 95% confidence band, indicated by shading. Source data are provided as a Source Data file.

## A CoDA meta-analysis of oligo- and pauci-mannose *N*-glycans in cancer

Previous results have established the robustness and sensitivity of our approach to glycomics data analysis. Next, we wanted to combine this with the established framework of engaging in meta-analyses, formally combining results across studies for robust conclusions, to showcase the biological implications of these analysis capabilities. For this, we used an existing set of comparative glycomics datasets investigating the relative abundance of oligo- and pauci-mannose *N*-glycans across different types of cancer[4,33], comparing healthy tissue and tumor samples (total $N = 194$ across nine datasets). While, for CLR-transformed data, this resulted in a number of structures that exhibited significant dysregulation across studies (Supplementary Data 11), other structures exhibited more modest or inconclusive effects (Fig. 5a).

One goal of such a meta-analysis is to identify potential biomarkers, or molecules to target for imaging and other endeavors. Hence, identifying the decrease of a structure such as Manα1-3(Manα1-6)Manα1-6(Manα1-3)Manβ1-4GlcNAcβ1-4GlcNAc in cancer ($d_{combined} = -0.5$, $p_{adj} = 0.0004$) could be viewed as an actionable finding. Yet this translation of relative decrease into absolute decrease pre-supposes equal scale between conditions. We thus repeated this analysis with an informed-scale model based on the ion intensities and noticed that, on average, the total glycan signal increased in cancer (average scale ratio: 1.34). Factoring in this scale difference led to much stronger findings (Fig. 5b, Supplementary Data 12) that were, on average, indicating an increase in most oligo- and pauci-mannose *N*-glycans in cancer, including the above-mentioned structure ($d_{combined} = 1.31$, $p_{adj} = 6.2 \times 10^{-12}$). Hence, analyzing the data without scale would have erroneously resulted in the conclusion that some of these structures decrease in absolute abundance between conditions, whereas the opposite seems to be true.

On the motif level, we observe similar trends (Fig. 5c, d, Supplementary Data 13 and 14), in that including an informed-scale model improves sensitivity and provides a better indication of the actual biological changes between conditions. In the case of terminal Manα1-2 epitopes (Fig. 5c, d), we for instance only report an absolute decrease in lung cancer and chronic lymphocytic leukemia, after factoring in scale differences between conditions. Findings from these methods could then provide a more solid foundation for (i) general-purpose biomarkers as well as (ii) specific markers that can be accurately tracked with methods that are sensitive to absolute expression levels (e.g., staining).

Reasoning that the combination of CLR and an informed-scale model do, in fact, more accurately identify biomarker candidates from glycomics data, we then engaged in a biomarker analysis via the ROC curve analysis workflow we developed herein. This indeed resulted in better predictive performance in a pan-cancer setting for the same structures (Fig. 5e, f; AUC 0.69 vs 0.81). We also noted that oligo- and pauci-mannose *N*-glycans in cancer exhibited a significantly higher alpha diversity (Fig. 5g; number of unique expressed structures; $p = 0.002$), again emphasizing the potential for cancer-relevant structures.

Lastly, since we had access to both *N*- and *O*-glycomics data of one cancer dataset, prostate cancer[34] ($N = 55$), we also performed a cross-class correlation analysis of CLR-transformed data (Fig. 5h). A positive correlation of fucose-containing glycans in both classes that we identified here could indicate a co-regulation across classes in prostate cancer, especially when contrasted to the lack of cross-class correlation of fucosylation during macrophage differentiation (Fig. 4e). Further, the neutral extension of core 2 structures was negatively correlated with oligo-mannose structures, while *O*-linked sialylation correlated positively with oligo-mannose structures. We again would like to note that these cross-class dependencies are crucial for interpretation, as an analysis that only factored in prostate cancer *N*-glycans might ascribe phenotypic effects to increases in oligo-mannose structures, while this is potentially confounded by concomitant increases in *O*-glycan sialylation. We thus urge researchers to factor in the entire glycome, which can be analyzed by methods such as the one presented here. We leave this section with a firm conviction that glycome analysis needs to be (i) statistically sound, (ii) robust yet sensitive, and (iii) integrating information across glycan classes, in order to be useful in the laboratory as well as, eventually, in the clinic.

## Discussion

Ignoring the compositional nature of data, in any systems biology discipline, is not an option. Intolerably high false-positive rates, analyses being dominated by the behavior of a few highly abundant glycans, or conclusions of differential expression that are the opposite of the biological reality—glycomics data need CoDA, if its results should serve as a solid foundation for the glycosciences. We show here that all these challenges are solvable and present a whole analysis suite to engage in various CoDA-powered glycomics analysis methods, that can yield robust biological insights and result in more actionable findings, as the indication of changes in absolute number of glycans across conditions aligns more closely with follow-up methods. An integration of all these state-of-the-art workflows into the easily accessible glycowork software platform[20] will ensure both compatibility with existing workflows and long-term support for these methods. While we show that these workflows can be applied to glycoproteomics data in principle, we caution that a proper workflow in future work would ideally need to account for the nested nature of glycoproteins, glycosites, and glycoforms.

We are especially enthusiastic about steps into rigorous analysis of multi-omics data that include glycomics data; facilitated by our flexible *get_SparCC* function, which is capable of comparing glycomics and non-glycomics datasets. Since this analysis modality still retains the flexibility to readily analyze glycan data on the motif level, identifying microbe-motif associations or gene-motif associations becomes almost trivially easy. We expect this to broaden the appeal of, and insight into, glycans as an explanatory modality across systems biology data types, or find application in evolutionary analyses comparing groups of species[35,36]. We also note that our treatment of other systems biology data (e.g., metagenomics or transcriptomics) as compositional data in this method is more appropriate than most existing approaches that are geared specifically toward analyzing those data types. Additionally, methods to measure several glycan classes from the same

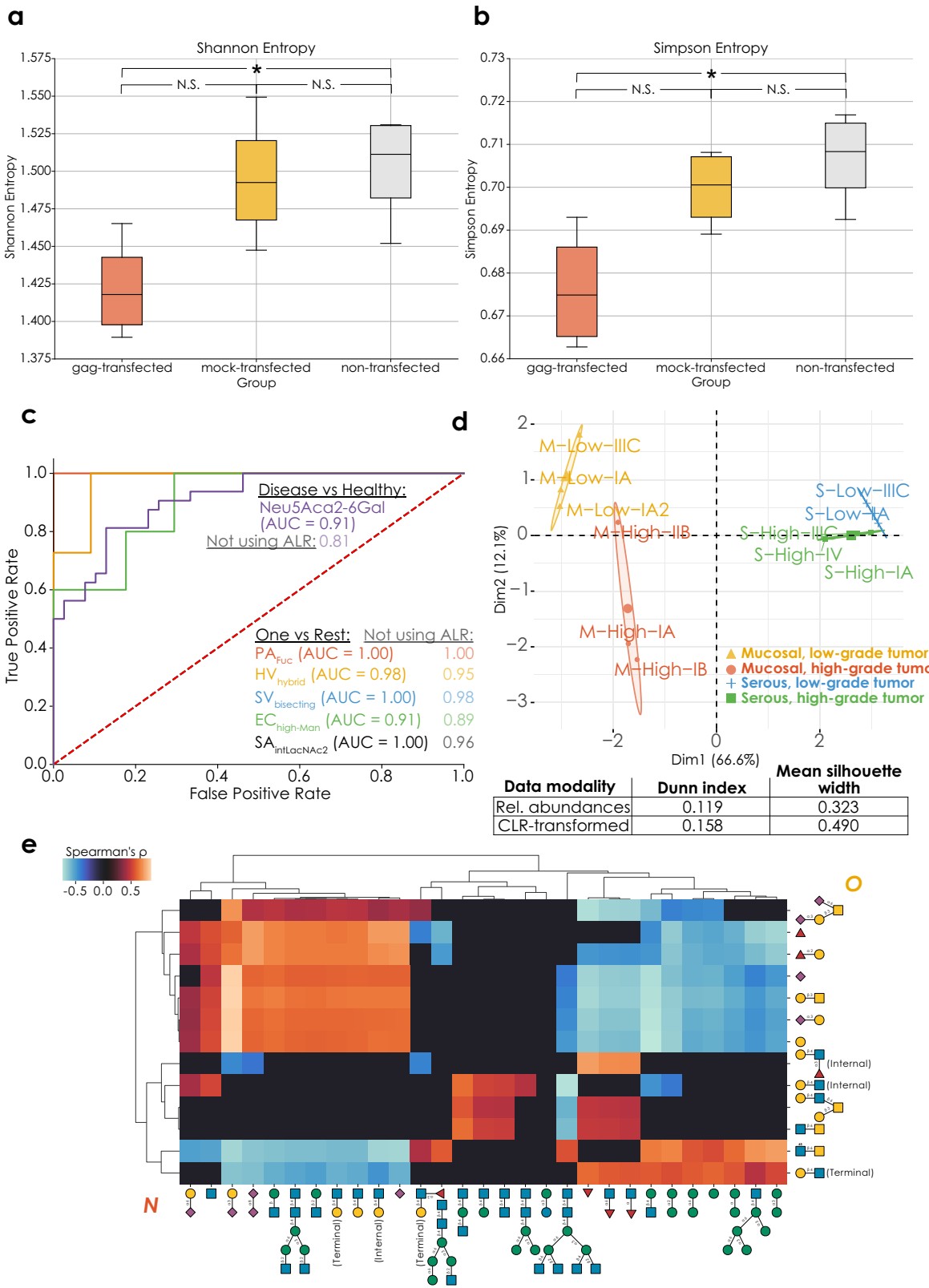

samples[37] also produce data that can be readily analyzed with this approach.

One source of bias that is currently not explicitly and routinely included in glycomics data analysis is technical variation, as measuring the same sample twice would likely result in slightly different values. One solution could be to include technical replicates in analyses, yet this information is rarely available. A microbiome differential abundance method, ALDEx2[9], addresses this by simulating technical

variation via Monte Carlo simulations of multinomial Dirichlet distributions, seeded by the actual relative abundances as concentration parameters. Then, each "instance" can be statistically tested and the test results can be averaged for a more robust inference. It is known, however, that this process is severely detrimental to the sensitivity of detecting effects for lowly abundant features[9] (i.e., most glycans) and is thus unsuitable for routine usage in glycomics. Instead, the reportedly more sensitive Dirichlet-multinomial modeling[38] could be a

**Fig. 4 | Specialized analysis modalities for comparative glycomics data.** Significant differences ($p < 0.072$, adjusted for a sample size of 8, see "Statistical analysis" section) between the *O*-glycome alpha diversities (**a**: Shannon entropy, $p = 0.011$; **b**: Simpson entropy, $p = 0.016$) of HEK293 cells transfected with HIV-1 *gag* and untransfected controls[30] ($n = 8$), determined via two-tailed Welch's *t*-tests with Benjamini–Hochberg correction. Data are depicted as mean values, with box edges indicating quartiles, and whiskers indicating the remaining data distribution up to the 95% confidence interval. **c** Identifying specific serum biomarkers for bacterial infection. Using the *get_roc* function (glycowork, version 1.3) on an ALR-transformed serum *N*-glycomics dataset with an informed-scale model, we obtained the best motif that separated healthy and bacteria-containing samples (total $N = 71$). We then, for each infecting pathogen, repeated this process as a One-Vs-Rest problem to identify the best motif that allowed us to separate the infecting pathogen from all other classes (other pathogens and healthy donors). Performances of this workflow without ALR transformation are also shown for comparison. PA *Pseudomonas aeruginosa*, HV Healthy volunteers, SV *Streptococcus viridans*, EC *Escherichia coli*, SA *Staphylococcus aureus*, Hybrid hybrid *N*-glycans, bisecting bisecting GlcNAc, high-Man high-mannose; intLacNAc2 internal LacNAc type 2. Area-under-the-curve (AUC) values are shown for all markers and a dashed line represents random assignment. **d** Beta-biodiversity clusters ovarian cancer types. We used a distance matrix from *get_biodiversity* (glycowork, version 1.3) of ovarian cancer *O*-glycomes[32] based on Aitchison distance for a principal component analysis (PCA), in which the different types and grades were well separated, indicated by colored clusters. The Eigenplot displays PCA clustering of distances from CLR-transformed data, with clustering metrics describing the comparison with PCA clustering of un-transformed data presented in an associated table. **e** Significant correlations ($p < 0.048$, adjusted for a sample size of 31) between CLR-transformed *N*- and *O*-glycomics data during differentiation from monocytes to macrophages[29], derived from the *get_SparCC* function (glycowork, version 1.3). Source data are provided as a Source Data file.

promising future avenue for further improving comparative glycomics analysis, yet would require major efforts to develop within Python. Other sources of variation when comparing glycomics data, such as incomplete labeling/reducing of glycans, comparing samples measured at different times, or sample degradation, have also been discussed before[39] and are, at least to some extent, mitigated by the inclusion of scale uncertainty.

We also caution that, while the inclusion of an informed-scale model can rectify misapprehensions of changes in conditions, it neither implies a specific mechanism nor suggests a causal relationship to the disease. Thus, if the total glycan signal increases in, e.g., cancer, several possible scenarios may be possible, such as changes in regulation (e.g., increasing glycosyltransferase expression) or distributional shifts (e.g., increasing expression of proteins that carry more glycans). Additionally, it is known that the exact quantification of glycans in a sample depends on the isolation strategy[40], which not only means that the experimental error estimate is crucial for robustness but also that this can be improved upon experimental advances.

We are convinced that the incorporation of CoDA into comparative glycomics analysis is a needed and especially timely one, as the unacknowledged bias present in current analysis methods is known to paradoxically increase false-positive rates with increasing sample size[18,19,41]. Others have noted previously that the rather modest sample sizes in systems biology fields, such as glycomics, have unexpectedly protected researchers from too many spurious findings[18,19,41]. As comparative glycomics is on the cusp of increasing data collection and analysis throughput[42,43], it becomes crucial to mitigate this unacknowledged bias, which is for instance achieved by accounting for scale uncertainty[18,19]. We thus envision that by making these cutting-edge statistical approaches to comparative glycomics data analysis easily available to the glycoscience community, the quality of results and conclusions will be raised, yielding a firmer understanding of glycans and their roles in biology.

## Methods

### Glycomics data simulation
Simulating glycomics data, such as to investigate the effect of sample size on type 1 error or sensitivity, largely followed the procedure we established previously[5]. Briefly, we used original relative abundances from carefully selected glycomics datasets as concentration parameters to sample replicates from a Dirichlet distribution that were paired with random glycans from SugarBase[20]. Known effects herein were introduced by scaling the concentration parameters of sialylated and fucosylated glycans up and down, respectively.

### Glycomics data generation
Standards included mammalian-type *N*-glycans (0.1 mg; NG-CM-051-M, NG-HM-001-OH, NG-CM-005-OH, NG-CM-054-OH, NG-CM-051-OH, all from NatGlycan) that were chemically reduced by dissolving them in 50 µL of 0.5 M NaBH₄ and 20 mM NaOH overnight at 50 °C. All standards were stored at 1 nmol/µL after reduction. For all samples, reduced glycan standards were mixed with 20 µL of released *O*-glycans from porcine gastric mucin (Type III PGM, Sigma-Aldrich, St. Louis, MI, USA). NG-HM-001-OH, NG-CM-005-OH, and NG-CM-051-OH were added to a final concentration of 2 pmol/µL for all samples, while both NG-CM-051-M and NG-CM-054-OH were added to final concentrations of 2, 6, 12, and 18 pmol/µL in triplicates.

Then, all samples ($N = 12$, with $n = 3$ biological replicates of each of the four conditions) were analyzed by liquid chromatography-electrospray ionization tandem mass spectrometry (LC-ESI-MS/MS), similar to our earlier work[36]. For glycan isomer separation, we used a porous graphite column (10 cm × 250 µm, packed in-house with 5 µm particles from Hypercarb, Thermo-Hypersil, Runcorn, UK). We injected 2 µL for all samples. Then, glycans were eluted with an acetonitrile gradient (Buffer A, 10 mM ammonium bicarbonate; Buffer B, 10 mM ammonium bicarbonate in 80% acetonitrile). We eluted the gradient (0–45% Buffer B) for 46 min, followed by washing with 100% Buffer B, and equilibrated with Buffer A for the next 24 min. We then used a 30 cm × 50 µm i.d. fused silica capillary as a transfer line to the ion source.

We analyzed all samples in triplicate in negative mode, using compressed air as the nebulizer gas, on a linear ion trap mass spectrometer (LTQ, Thermo Electron, San José, CA), with an IonMax ESI source and a stainless-steel needle at −3.5 kV. We maintained the heated capillary at 270 °C and −50 kV. We then performed a full scan ($m/z$ 380–2000, two micro scans, maximum 100 ms, target value of 30,000) in data-dependent acquisition mode. For data acquisition, we used Xcalibur (Version 2.0.7, Thermo Scientific) and for the quantification of glycans via their ion intensity we used Progenesis QI (Nonlinear Dynamics, Waters Corp., Milford, MA, USA). The LC-ESI-MS/MS raw data have been deposited in GlycoPOST under the accession number GPST000487.

### Datasets
Datasets analyzed in this work have been sourced from the academic literature, by processing supplementary tables. Where possible, glycan sequences were converted into IUPAC-condensed nomenclature, either manually or using the *canonicalize_iupac* function within glycowork, followed by manual inspection. Quantitative values were usually recorded as relative abundances, summing up to 100%, where this was not already performed. If the original authors imputed zeroes with a small constant (e.g., 0.1), then we reverted this back to zero. Otherwise, no alteration of the original data was performed. The processed datasets can be found within glycowork (version 1.3+, available via the *glycomics_data_loader*) and/or the supplementary tables in this work.

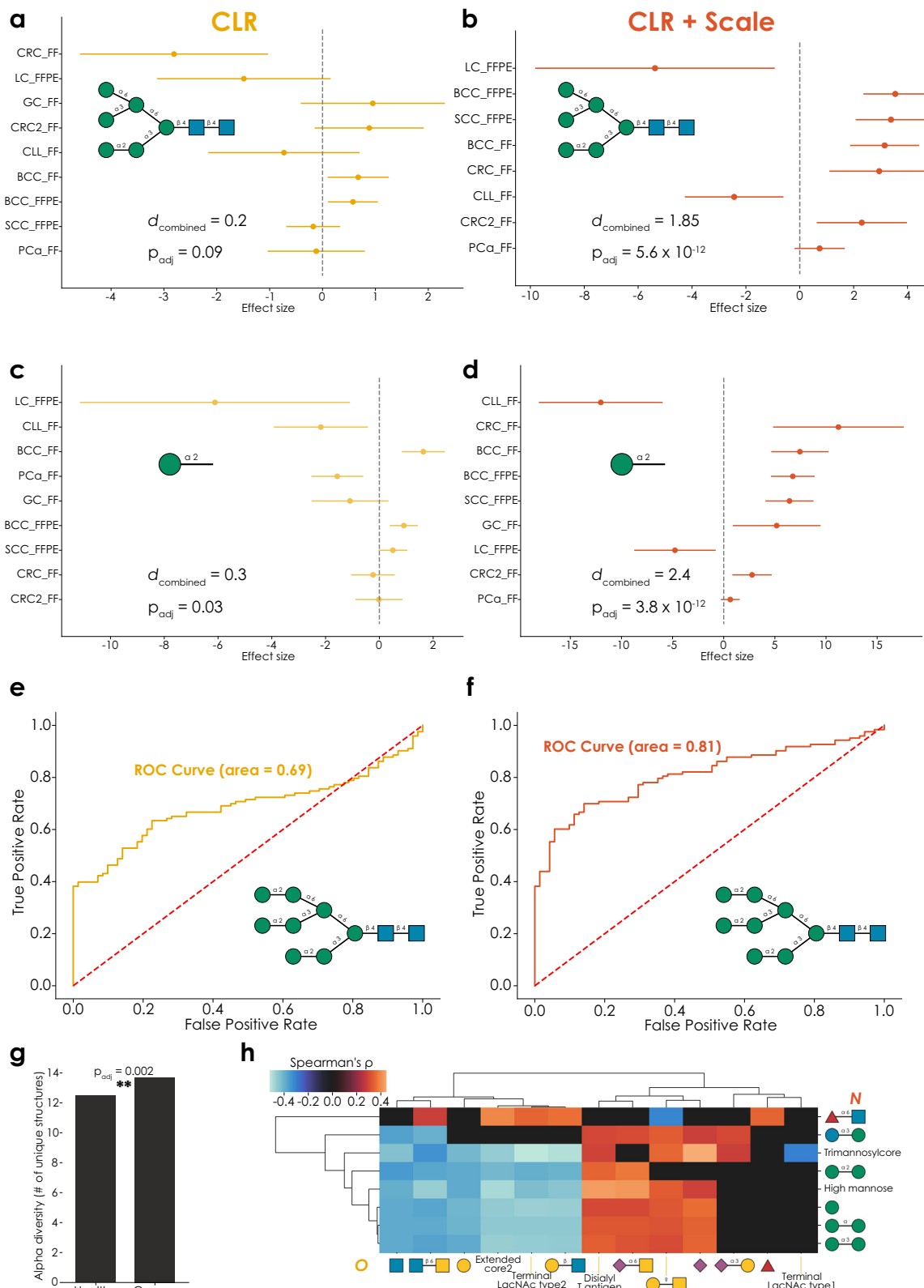

## Data processing

As a general processing step in all functions, glycans with zero values in all samples were dropped, followed by outlier treatment via Winsorization[44]. Briefly, this procedure replaced the 5% most extreme values on either end by the data-dependent values of the 5th and 95th percentile, respectively, which has been shown to be effective across all sample sizes to reduce the impact of outliers on statistical testing[44].

Next, missing values were assumed to be missing at random and imputed via an iterative application of a MissForest algorithm[5,45] implemented within glycowork, starting with median values as placeholders and proceeding for up to five iterations or convergence. An exception to this was made when all samples of one condition exhibited zero expression of a glycan. In such cases, zeroes were viewed as not missing at random and were replaced with an arbitrarily small

**Fig. 5 | A compositional data meta-analysis of oligo- and pauci-mannose *N*-glycans in cancer.** Both full structures (**a**, **b**) and motifs (**c**, **d**) are consistently dysregulated across cancer types in a fixed-effects meta-analysis of CLR-transformed relative abundances (using the *get_meta_analysis* function within glycowork, version 1.3), using a single two-tailed *t*-test of the combined effect size. Adding an informed-scale model based on the ion intensities (**b**, **d**) improves analytical sensitivity and reveals an absolute increase of most considered structures and motifs in cancer. Total *n* = 194 and combined effect sizes are shown as Cohen's *d*. All results can be found in Supplementary Data 11–14. A pairwise analysis of CLR-transformed data from healthy samples and cancer tissues reveals structures of moderate predictiveness (**e**; AUC = 0.69), which is substantially improved by the informed-scale

model (**f**; AUC = 0.81). ROC curves were analyzed and generated with the *get_roc* function (glycowork, version 1.3). **g** In a pan-cancer analysis of oligo- and pauci-mannose *N*-glycans, cancer samples exhibited a significantly greater alpha diversity of these structures (*p* = 0.002), measured as the number of expressed unique structures per sample, calculated with the *get_biodiversity* function (glycowork, version 1.3), using a single two-tailed Welch's *t*-test. **h** Correlating CLR-transformed *N*-glycomics and *O*-glycomics data from the same prostate cancer patients in different stages reveals substantial cross-class regulation. Only significant correlations, derived from the *get_SparCC* function (glycowork, version 1.3), are shown as Spearman's rho, with everything else set to zero. **\*\****p* < 0.01. Source data are provided as a Source Data file.

constant ($10^{-5}$) to facilitate the transformations below. Thereafter, each sample was re-normalized to a total sum of 100. All data processing steps (and analyses below) have been entirely executed within glycowork (version 1.3).

### Transforming compositional data

While glycowork contains a substantial amount of functionality targeted at the monosaccharide compositions (e.g., mass calculation or structure mapping), we here focus on the statistical compositions mentioned above. By default, data were transformed from the Aitchison simplex to real space via a CLR transformation. This meant subtracting the log2-transformed geometric mean from the log2-transformed data. Further, the geometric mean here was only formed from the non-zero elements of the data. In the case of an applied scale uncertainty model[14,18,19], this log-transformed geometric mean was then modified by drawing samples from a normal distribution, with the log-transformed geometric mean as its center and an $\gamma$ parameter that controlled its variance ($\gamma^2$). The default $\gamma$ value in all glycowork functions is set as 0.1, to always account for at least some measure of scale uncertainty.

Alternatively, we also offer the possibility for an ALR transformation, which is the default transformation for *N*-glycomics datasets with >50 identified structures. Here, the ratio of all glycans with a reference glycan is formed after log2-transformation. To identify an appropriate reference glycan that fulfills the requirements of CoDA, we rank all possible glycans by a custom score that multiplies Procrustes correlation by inverse glycan variance across conditions[46]. Here, the Procrustes correlation measures how well the ALR transformation using a candidate glycan re-captures the geometry achieved by a CLR transformation of the same data. The glycan with the highest Procrustes correlation and lowest variance is then chosen as the reference, and hence excluded from analysis. If that glycan exhibited a Procrustes correlation of below 0.9, or a between-group variance of above 0.1, ALR was abandoned and data were transformed via CLR. We note that neither CLR nor ALR are zero-bounded, so negative values are to be expected in the output and simply translate to small(er) values.

Finally, if users have knowledge about the actual scale difference (e.g., total number of glycans 20% lower in the treatment condition), this information can be used for an informed-scale model, akin to previous work[14], which transforms the meaning of $\gamma^2$ into an estimate of the experimental error in determining that scale difference. For binary comparisons, we used the ratio of the group scales, whereas a dictionary of the scales, expressed as ratios to the lowest scale, was used in multi-group settings. When using ion intensities for an informed-scale model, we used the mean of the sums of ion intensities for each sample across a condition as an indicator for its average total glycan amount.

### Differential expression analysis

The key function for pairwise differential expression in glycowork is *get_differential_expression*, together with *get_glycanova* for ANOVA-based set-ups.

If the analysis was performed on the motif level, CLR/ALR-transformed data were used for the *quantify_motifs* function in glycowork, which proportionally scales and sums abundances across different glycans exhibiting a certain motif/substructure. Motifs with redundant information were identified and handled via the *clean_up_heatmap* function in glycowork, which retains the largest motif of a redundant group. Otherwise, the abundances of duplicate sequences were averaged. Subsequently, all glycans/motifs with less than 2% overall variance were discarded. Mean abundances of glycans/motifs were calculated from the original relative abundances. Then, for each glycan or motif, the CLR/ALR-transformed values were tested for differential abundance (Welch's *t*-test or paired *t*-test), effect size (Cohen's $d/d_z$), heteroscedasticity (Levene's test), and equivalence (two one-sided *t*-tests).

By default, we use the two-stage Benjamini–Hochberg procedure for multiple testing correction[47]. Optionally, users can employ a grouped two-stage Benjamini–Hochberg procedure for increased sensitivity[48], in which glycans are grouped by biosynthetic similarity (e.g., core structures in *O*-glycans, high-mannose/complex/hybrid in *N*-glycans, etc.). Meaningful groupings are confirmed by comparing intra-group correlation to inter-group correlation via a linear mixed-effects model. Groupings are only employed if the intra-group correlation is three times larger than the inter-group correlation and larger than 0.1, otherwise a standard two-stage Benjamini–Hochberg procedure is executed. In either case, and also for the methods mentioned below, if the fraction of significant results ever exceeds 90%, a problem in the CLR/ALR transformation is assumed and, for the sake of robustness, multiple testing correction is switched to the conservative Bonferroni correction.

### Biodiversity analysis

Both alpha- and beta-diversity metrics can be mixed and matched (both on the sequence and motif level) in the *get_biodiversity* function in glycowork.

Alpha diversity metrics are designed to operate on count-based data and use relative abundances, while beta-diversity metrics require distances and thus use CLR/ALR-transformed data[6]. For alpha diversity, we used sequence richness, Shannon entropy, and Simpson entropy as metrics, which were then compared via two-tailed Welch's *t*-tests for significant group differences. If more than two groups were present, these tests were performed as ANOVAs on the respective alpha diversity metrics.

For beta diversity, we constructed a distance matrix based on Aitchison distance (Euclidean distance after CLR/ALR transformation into real space) of glycans/motifs and then performed an analysis of similarities (ANOSIM)[49] and a permutational multivariate analysis of variance (PERMANOVA)[50] on this distance matrix. Briefly, an ANOSIM compares between-group mean distance and within-group mean distance against the null hypothesis that they are equal, resulting in an *R* value. A PERMANOVA compares variances in distance in a between-group versus within-group setting, resulting in an *F*-statistic. In both methods, a *p* value is obtained by comparing the actual difference (or

F-statistic) against differences (or F-statistics) obtained by shuffling group labels in a large number of permutations ($n = 999$ in our case). Obtained $p$ values were then corrected for multiple testing via a two-stage Benjamini–Hochberg procedure.

## Cross-class associations

Using the *get_SparCC* function in glycowork, two systems biology datasets can be cross-correlated. If glycomics datasets are used, this can be performed at the whole-glycan or motif level. After processing, both datasets are CLR/ALR-transformed and a Spearman correlation matrix is calculated from the transformed values[21]. Pairwise correlations are tested for significance via two-tailed $t$-tests, corrected for multiple testing by a two-stage Benjamini–Hochberg procedure.

## Statistical analysis

For all analyses, a sample-size appropriate α level for statistical significance was chosen via Bayesian-aware alpha adjustment[51], to always obtain a Bayes factor of at least three for the threshold of statistical significance (*get_alphaN* in glycowork). For statistical analysis, this study used two-tailed Welch's $t$-test for univariate and Hotelling's $T^2$ test for multivariate comparisons. Differences in variance were tested by Levene's test. Pairwise post-hoc comparisons were done with Tukey's honestly significant difference test. All multiple testing corrections were done via the two-stage or grouped two-stage Benjamini–Hochberg procedure. Effect sizes were estimated via Cohen's $d/d_z$ for univariate and the Mahalanobis distance for multivariate comparisons. All statistical testing has been done in Python 3.11.3 using the glycowork package (version 1.3), the statsmodels package (version 0.14), and the scipy package (version 1.11). Friedmann analysis was conducted in GraphPad Prism (version 10). Data normalization and motif quantification were done with glycowork (version 1.3).

## Reporting summary

Further information on research design is available in the Nature Portfolio Reporting Summary linked to this article.

## Data availability

All data used in this article can either be found in supplementary tables or as stored datasets within glycowork. Unless otherwise stated, all data supporting the results of this study can be found in the article, supplementary, and source data files. Generated glycomics data for this article can be found on GlycoPOST, under the accession number GPST000487. Source data are provided with this paper.

## Code availability

Code and documentation are available via glycowork v1.3 [https://github.com/BojarLab/glycowork][52], which can also be accessed via Zenodo [https://zenodo.org/records/11543487].

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

## Acknowledgements

This work was supported by a Branco Weiss Fellowship—Society in Science awarded to D.B.; by the Knut and Alice Wallenberg Foundation; and the University of Gothenburg, Sweden. S.C. was supported by an International Macquarie Research Excellence Scholarship (iMQRES). MTA is the recipient of an ARC Future Fellowship from the Australian Research Council (FT210100455). We thank SciLifeLab and BioMS funded by the Swedish Research Council for providing financial support to the Proteomics Core Facility, Sahlgrenska Academy.

## Author contributions

Conceptualization: D.B., funding acquisition: D.B., resources: D.B., M.T.A., S.C., software: A.R.B., D.B., J.L., supervision: D.B., M.T.A., visualization: A.R.B., D.B., J.L., writing—original draft preparation: A.R.B., D.B., J.L., writing—review & editing: A.R.B., D.B., J.L., M.T.A., S.C.

## Funding

## Competing interests

The authors declare no competing interests.
