## [Transparent Peer Review file · Nature Communications]

Compositional Data Analysis Enables Statistical Rigor in Comparative Glycomics

Corresponding Author: Dr Daniel Bojar

Version 0:

Reviewer comments:

Reviewer #1

(Remarks to the Author)

The manuscript presents a novel approach to comparative glycomics analysis using the center log-ratio (CLR) and additive log-ratio (ALR) transformations. The authors successfully demonstrated the superiority of these methods over traditional percent abundance normalization, addressing a long-standing need in the field. The CLR and ALR transformations offer a more robust and accurate quantitative analysis of glycomics data, potentially leading to significant advancements in research. However, the manuscript raises concerns regarding the potential discrepancies between CLR-based analysis and previously published data. While the authors attribute this to the limitations of percent abundance normalization, a more comprehensive comparison with a standard sample containing known glycan concentrations is necessary to validate the accuracy and reliability of the proposed methods.

The manuscript tries to have statical solutions and new options for analysis of glycomics data that produces composition analysis of glycan. Glycomics experiments are currently calculated based on relative % of each glycan. This procedure produces some high false positive rates mostly influenced by few high intensity glycans. They have introduced a new set of various CoDA- glycomics methods, that can help better calculations for large glycomics data sets. They have applied this CoDA analysis to number of publicly available data sets and redid the calculations to estimate the relative glycan amounts. The effort made is much needed for the field specially as more glycomics data is becoming available and more high throughput methods are being employed. The accuracy of data analysis in regard to statistical accuracy of the glycomics data for N and O-linked glycan, milk oligosaccharides, glycolipid analysis or glycoproteomic is critical. Their approach helps eliminate the problem of the current methods are where an increase in one part necessitates the decrease of another. Although their approach of the center log-ratio (CLR) and additive log-ratio (ALR) proposes to provide more accurate composition data in glycomics, the data are not really tested against absolute values. The methods they propose will be very useful for the glycoscience community specially the open-source glycowork Python package that comes with the manuscript. It will be best if they can add some text or additions to show what it would take for them to actually test their method with quantitative data maybe from NMR. If they can add few glycans and measure the actual amount of each and measure that by NMR and then measure a second set of samples with different glycan % that has also been verified by NMR and then compare their glycomics composition statistical analysis with just simple current relative % and determine the power of their applied CLR method

Comment 1 -

The manuscript's evaluation of the proposed CLR and ALR transformations would benefit significantly from the inclusion of a dataset containing absolute glycan quantification. While the chosen published datasets offer valuable insights, the lack of absolute values hinders a complete assessment of the method's accuracy. As suggested by Moh et al. (2015), incorporating a dataset with known absolute glycan concentrations would provide a more robust comparison and validate the method's ability to translate relative differences into accurate quantitative measurements.

Moh, E. S., Thaysen-Andersen, M., & Packer, N. H. (2015). Relative versus absolute quantitation in disease glycomics. *Proteomics. Clinical applications*, 9(3-4), 368–382. <https://doi.org/10.1002/prca.201400184>

Comment 2 -

While the database from PMID35112714 may have provided valuable insights for the initial analysis, it is essential to explain the specific reasons for its selection. If statistical data regarding glycome composition were not reported, it would be

helpful to elaborate on how the database's characteristics or the specific research questions addressed by the study justified its use. This would enhance the transparency of the methodology and strengthen the overall validity of the findings.

Comment 3-

In Table 1, the original finding column for reference PMID36952551 states that samples were stored at 20°C. However, the original article indicates a storage temperature of 22°C. Please verify and correct this.

Comment 4 -

Table 1 lists reference PMID27997589 as having 'no analysis of the glycomics data reported.' If this is indeed the case, it would be important to explain how the data from this study was incorporated into the present analysis. Please provide additional details on the specific methods used to extract or analyze the glycomics data from this reference or explain why it was included in the table if no analysis was conducted.

Comment 5 -

Supplementary Figure 1 compares quantitative glycopeptides using log₂-transformed and ALR/CLR-transformed relative abundances. While the data reveals a set of glycopeptides that became significant with the ALR/CLR method, a more detailed explanation of the underlying reasons for this difference is warranted. Exploring potential factors such as the glycopeptides' inherent characteristics, the methods' sensitivity to specific glycan features, or the impact of normalization techniques would provide valuable insights into the observed discrepancies.

The manuscript introduced the center log-ratio (CLR) and additive log-ratio (ALR) transformations approach to analyze comparative glycomics data. The author analyzed the publicly available glycomics datasets and demonstrated the pitfall of the traditional statistical analyses, which are based on percent abundance normalization as a sum of the whole glycan. CLR and ALR methods proved to be better statistical than the Aitchison simplex method. The glycomics community has been needing a better and more robust quantitative method for glycomics for a long time. The addition of this method for the glycomics analysis will be a major add-on to the subject area.

However, I have a few concerns. One is that CLR-based analysis shows different outcomes from the already published data. Some potential changes in the glycans were nullified, and other glycans emerged as significant changes. This will eventually redirect the research outcome and future directions.

It is known that data % abundance normalization is biased when introducing artificial changes in abundance value as it is relative to the other part of the data. Thus, it is very important to compare this proposed method and the Aitchison simplex method to a standard sample where known concentration of glycan was endogenously expressed or spiked-in of a different class of the glycans like High-mannose, complex/hybrid, sialylated, and both fucosylated and sialylated type of glycans at different concentration and assess the performance of quantitative analysis.

(Remarks on code availability)

Reviewer #3

(Remarks to the Author)

Bennett and colleagues demonstrate the application of CLR transformation instead of normalized glycan ratios, which result in compositional data. The authors have made functions available through their Glycowork Python package, which is a highly useful resource for the glycomics field.

The main concern currently is whether the presented work is novel enough for publication in Nature Communications. However, this transformation has been employed in other omics fields before. Furthermore, transformations in glycomics were also discussed and evaluated in this article: Uh, Hae-Won, et al. "Choosing proper normalization is essential for discovery of sparse glycan biomarkers." *Molecular omics* 16.3 (2020).

There is clearly a need for better statistical tools in the glycomics field, as there are likely many false positives and negatives in published data. However, there are many other experimental and technical factors influencing the outcomes of glycomics data. The authors should elaborate on these points.

Minor points

-> That's for an abstract difficult to follow, from where does this number come from? "or routine false-positive rates of >25% for differential abundance"

Line 103: "Additionally, as will be shown, the total number of molecules is rarely invariant across conditions, further distorting the obtained relative abundances and their differences across conditions."

-> This is exactly the scenario that would occur when the occupation of a glycosylation site increases from 1% to 100%. The additional glycans will decrease the relative abundance of all other glycans. Why do the authors not only focus on glycoproteomics data? Isn't it most important to know which protein and which glycosylation site the changes occur at?

Line 73 and Table 1: "When applied to a range of glycomics datasets, we show that differential expression analysis using this enhanced CLR/ALR transformation can reveal biological insights."

-> These findings, are difficult to verify. How can we be sure that these are genuine biological insights and not (again) artifacts? This demonstrates the complexity of the field, as small changes in the workflow may provide very different outputs.

Reference 6: The referenced article by Gloor and colleagues is a review article, and the methods have been applied earlier

in different fields. This should be correctly referenced and represented.

(Remarks on code availability)

Reviewer #4

(Remarks to the Author)

The authors present a good solution to a problem that I've run afoul of before without an obvious right way to solve it. This is an important method to share with the community, and to help to draw attention to this recurring issue with statistical analysis.

Major comments

1. There are a lot of places where the mathematical nomenclature uses terms that have a distinct meaning in mass spectrometry-based glycomics analysis. If you want to use the nomenclature that is consistent with CoDA, please make your definitions clear early on.
 - a. Composition analysis may refer to either the analysis of the number of each of a thing in a population or an analysis of glycans by monosaccharide composition, not structure. Your target audience may not understand which you mean, making it harder for the importance of your methods to come across. It may be that `glycowork` does not have a concept of an unstructured bag of monosaccharides, which might also be clarified.
 - b. Relative abundance, in the context of a population is the proportion of different components of that population, but in mass spectrometry, relative abundance refers to the ion intensity measured by the instrument, either as a point value in a spectrum or integrated over time. The latter is what most MS-based quantitative experiments use to quantify glycans unless isotopically labeled reagents like tandem mass tags (TMT) are used, which is a measure of absolute abundance because it lets us effectively ignore ionization efficiency by quantifying reporter ions from fragmentation instead of the precursor.
 - c. The term "feature" is used interchangeably with "glycan", but not all features are glycans, and some features are backed by multiple glycans. Please use explicit language to make it clear which you are referring to.
2. The narrative is fragmented as it tries to touch on at least five datasets with little explanation about what the objective of each study was, which in turn makes it hard to understand how or why the metric of success changes. Additionally, after the first comparison shown in Figure 2, the reader is never shown a head-to-head comparison of CoDA to demonstrate how it helps on experimental data with real world noise characteristics.
3. The scale uncertainty components of `glycowork.glycan_data.stats.clr_transformation` and `.alr_transformation` invoke the global NumPy RNG without any management of the random state or random seed, which means that unless one seeds the RNG immediately prior to invoking a `glycowork` function that calls one of them, the result is non-deterministic. How much do the results shown in the paper vary over multiple invocations of these functions?
4. The supplementary tables show only the final tables of differences. No "Source Data" tables showing the data used to produce the plots in the main text, which makes reproducing the analysis even more problematic. It is not obvious if the MissForest algorithm mentioned in the methods is even a component of `glycowork`.

Minor comments

5. The repeated expression of "glycan or glycopeptide" uses a lot of space for not a lot of added value after this is stated for the fourth time. A glycosite is a glycome in microcosm, but it has its own complications. A glycosite spanned by a fully and partially mis-cleaved peptide bearing the same glycan are not truly additive because of differences in peptide sequence ionization efficiency. Also make sure to plug glycolipids if it is suitable for them as well.
6. The comments regarding ionization efficiency and MS quantification gives the impression that spectrum counting a common form of quantification. I doubt it is reasonable to claim so in isolation.
7. The discussion of informed scale models mentions using the ion intensities as a the value of ζ , but it isn't specified whether $\text{mean}(\text{ion})_{\text{glycans}}_{\text{condition}}$ is the average of the sum of all glycans in each sample in the condition or some other function. I think the methods section later indicates this is a geometric mean, but that is a source of confusion.

Thank you,
Joshua Klein

(Remarks on code availability)

As described in the main commentary, the code as written uses a global RNG that would have to be controlled by the caller to make any function invocation reproducible. Without the program that was used to produce the figures in the paper, along with their final conclusions it would not be possible to determine if a random seed was used.

The code is readable and documented, if difficult to navigate due to the dozens of free functions per file as encouraged by the nbdev programming style. The docstrings are written in a format that is neither the official rST notation, NumPy, nor Google style syntax, instead opting to render them as rST tables, but most functions are documented.

The rest of the codebase is functionally solid. Minor quibbles about internal representation types being loosely defined strings and DataFrames aside.

The git repository should not track the top-level `build` or `_proc` directories, and should purge the `__pycache__` directories as well, but this is minor. It may be desirable to replace several of the larger static data files with compressed formats that

reduce the overhead when transporting them, e.g. Parquet or compressing the csv files, and to gzip the pickle files.

Version 1:

Reviewer comments:

Reviewer #1

(Remarks to the Author)

Here is our comment:

The authors have addressed our major concern by demonstrating the CoDA-based method with absolute glycan concentrations. They have included a new experiment with glycan standards and incorporated the standard glycan comparison data into Figure 1b of the revised manuscript. Additionally, they have provided a detailed explanation of the results.

The authors have effectively addressed all of our comments and concerns. We appreciate their addition of the constant 2 pmol concentration of standards 1 and 2 (glycan m/z: 893.4 and 941.39) in all samples, while standards 3 and 4 (glycan m/z: 1009.94 and 1111.46) are in increasing order. The pattern of standard glycan concentrations and the skewed nature of relative abundance are well-captured in the ALR-transformed data (Figure 1b). However, we note the negative value for glycan 941.39 in the ALR-transformed data (Figure 1b and Supplementary Table 1). We request an explanation for this negative trend and guidance on how to compensate or adjust for this observation in the analysis to accurately express glycan abundance in real datasets.

Additionally, we suggest clarifying the bar color code in Figure 1b, including the gray color.

Overall, the revised manuscript, with the inclusion of the absolute concentration experiment for glycan standards, has significantly improved the validity and readability of the CoDA-based method for quantitative analysis of glycomics data compared to % relative abundance. This revised manuscript is well well-suited for journal Nature Communications.

(Remarks on code availability)

Reviewer #2

(Remarks to the Author)

(Remarks on code availability)

Reviewer #3

(Remarks to the Author)

I am satisfied with the authors' response. I have no additional comments.

(Remarks on code availability)

Reviewer #4

(Remarks to the Author)

The authors' revisions to the manuscript addressed all of my concerns, and provided pointers to where to track down specific technical details of the described experiments.

(Remarks on code availability)

A dedicated RNG is now created and seeded on module import, but it is still not controlled at the call site of `clr_transform`` and friends, and worse, it is now even harder for the caller to control! As the authors addressed my concerns about the stability of the method w.r.t. the random element in the revision, the following is a suggestion for code quality, not a requirement for revision.

To control it, the user must explicitly write `glycoworks.glycan_data.stats.rng = np.random.default_rng(...)``. Instead, the caller should be able to pass a `RandomState`/RNG-seed-like` value as an optional parameter to produce an RNG for the function and all its callees to use, falling back to some default seed/shared RNG state when the argument is absent. This is a pervasive pattern in the `scikit-learn`` library.

This does involve piping a new parameter throughout the call-graph which the authors may find too odious to write, in which case I suggest adding a note to the documentation about the shared RNG, and adding a top-level function that can update the shared RNG without breaking encapsulation.

We thank all reviewers for their insightful comments and suggestions for improvement. We have fully addressed these comments in our substantially revised manuscript by engaging in new experiments and analyses, extensive text modifications and additions, as well as additions to our supplementary tables. In summary, we have (i) performed new glycomics experiments to measure glycan standards in different known concentrations and validate our CoDA-based workflow (**new Fig. 1b**), (ii) engaged in a new analysis to show that our results are qualitatively and quantitatively reproducible (**new Supplementary Fig. 4**), (iii) further enhanced reproducibility by fixing all random seeds and generally cleaned up the glycowork repository by removing unnecessary files, (iv) added an additional head-to-head comparison of traditional vs CoDA-based analysis (**revised Fig. 4c**), (v) added more context to the analyzed studies and expanded our discussion, (vi) enhanced text clarity by defining relevant terms and using them consistently, as well as other improvements detailed throughout this document. Changes in the manuscript and point-by-point responses here are colored in blue. We believe that these changes have substantially improved our manuscript, contextualized our findings, and will allow readers to better evaluate our analyses and findings.

Reviewer #1 (Remarks to the Author):

The manuscript presents a novel approach to comparative glycomics analysis using the center log-ratio (CLR) and additive log-ratio (ALR) transformations. The authors successfully demonstrated the superiority of these methods over traditional percent abundance normalization, addressing a long-standing need in the field. The CLR and ALR transformations offer a more robust and accurate quantitative analysis of glycomics data, potentially leading to significant advancements in research. However, the manuscript raises concerns regarding the potential discrepancies between CLR-based analysis and previously published data. While the authors attribute this to the limitations of percent abundance normalization, a more comprehensive comparison with a standard sample containing known glycan concentrations is necessary to validate the accuracy and reliability of the proposed methods.

The manuscripts tries to have statical solutions and new options for analysis of glycomics data that produces composition analysis of glycan. Glycomics experiments are currently calculated based on relative % of each glycan. This procedure produces some high false positive rates mostly influenced by few high intensity glycans. They have introduced a new set of various CoDA- glycomics methods, that can help better calculations for large glycomics data sets. They have applied this CoDA analysis to number of publicly available data sets and redid the calculations to estimate the relative glycan amounts. The effort made is much needed for the field specially as more glycomics data is becoming available and more high throughput methods are being employed. The accuracy of data analysis in regard to statistical accuracy of the glycomics data for N and O-linked glycan, milk oligosaccharides, glycolipid analysis or glycoproteomic is critical. Their approach helps eliminate the problem of the current methods are where an increase in one part necessitates the decrease of another. Although their

approach of the center log-ratio (CLR) and additive log-ratio (ALR) proposes to provide more accurate composition data in glycomics, the data are not really tested against absolute values. The methods they propose will be very useful for the glycoscience community specially the open-source glycowork Python package that comes with the manuscript.

It will be best if they can add some text or additions to show what it would take for them to actually test their method with quantitative data maybe from NMR. If they can add few glycans and measure the actual amount of each and measure that by NMR and then measure a second set of samples with different glycan % that has also been verified by NMR and then compare their glycomics composition statistical analysis with just simple current relative % and determine the power of their applied CLR method

We thank the reviewer for their careful assessment of our work. We address each specific comment below but would like to summarize that we have engaged in new experiments, to demonstrate the utility of our approach with known absolute glycan concentrations, as well as numerous text additions and clarifications to better contextualize our work and make it clearer for readers.

Comment 1 -

The manuscript's evaluation of the proposed CLR and ALR transformations would benefit significantly from the inclusion of a dataset containing absolute glycan quantification. While the chosen published datasets offer valuable insights, the lack of absolute values hinders a complete assessment of the method's accuracy. As suggested by Moh et al. (2015), incorporating a dataset with known absolute glycan concentrations would provide a more robust comparison and validate the method's ability to translate relative differences into accurate quantitative measurements.

Moh, E. S., Thaysen-Andersen, M., & Packer, N. H. (2015). Relative versus absolute quantitation in disease glycomics. *Proteomics. Clinical applications*, 9(3-4), 368–382. <https://doi.org/10.1002/prca.201400184>

We agree with the assessment of the reviewer and have added new experiments, in which we add glycan standards in known absolute concentrations (**new Fig. 1b**). We then used our herein developed methods to showcase that, while conventionally used analysis methods suffer from false-positives in such a scenario, our CoDA-based methodology correctly recaptures the known ground-truth effects. We thus are confident that we present a robust analysis platform that constitutes an improvement over existing comparative glycomics analysis workflows. We also now reference the indicated work in our revised manuscript.

Comment 2 -

While the database from PMID35112714 may have provided valuable insights for the initial analysis, it is essential to explain the specific reasons for its selection. If statistical data regarding glycome composition were not reported, it would be helpful to elaborate on how the database's characteristics or the specific research questions addressed by the study justified its use. This would enhance the transparency of the methodology and strengthen the overall validity of the findings.

Since PMID35112714 still has made qualitative conclusions about changes between conditions in their glycomics data, we would argue that, precisely because no statistical analysis has been reported for the indicated study, an evaluation with our new approach is important to assess the validity of the conclusions. In this particular case, we observe that, while we do confirm the initial conclusions (changes in glycosylation by transfection with HIV-1 Gag and mock plasmids), the change of *N*-glycans between non-transfected and mock-transfected cells does not reach statistical significance. This finding is at least not explicitly contained within PMID35112714 and thus showcases the importance of applying our new methodology to this case. We have added this consideration into our revised manuscript.

Comment 3-

In Table 1, the original finding column for reference PMID36952551 states that samples were stored at 20°C. However, the original article indicates a storage temperature of 22°C. Please verify and correct this.

We have corrected this in the revised manuscript.

Comment 4 -

Table 1 lists reference PMID27997589 as having 'no analysis of the glycomics data reported.' If this is indeed the case, it would be important to explain how the data from this study was incorporated into the present analysis. Please provide additional details on the specific methods used to extract or analyze the glycomics data from this reference or explain why it was included in the table if no analysis was conducted.

We thank the reviewer for pointing this out. What we intended to convey—and we have changed this now for the revised manuscript—was that no formal statistical analysis of the glycomics data was carried out in PMID27997589. However, this study still contains claims about effects in the different conditions (i.e., different tissues and different groups of gangliosides), which allows us to confirm and/or revise findings with a formal statistical analysis.

Comment 5 -

Supplementary Figure 1 compares quantitative glycopeptides using log₂-transformed and ALR/CLR-transformed relative abundances. While the data reveals a set of glycopeptides that became significant with the ALR/CLR method, a more detailed explanation of the underlying reasons for this difference is warranted. Exploring potential factors such as the glycopeptides' inherent characteristics, the methods' sensitivity to specific glycan features, or the impact of normalization techniques would provide valuable insights into the observed discrepancies.

This is an interesting thought and we have more closely investigated the results of our analysis in Supplementary Figure 1. Examining the milk glycoproteomics data (Supplementary Fig. 1a-b) revealed that many of the glycopeptides that are identified as differentially-expressed (DE) by CLR-transformation, but not log₂-transformation, have relatively small fold-changes. To illustrate this, the mean fold-change among DE glycopeptides identified by log₂-transformation was 9.11, while the mean among CLR-transformed DE glycopeptides was 2.24. We frequently observe that log₂-transformation captures the few DE glycopeptides with the greatest fold-change, or overall abundance. CLR-transformation, however, is also able to detect DE among relatively less-abundant glycopeptides, or those with smaller fold-changes.

In the datasets we have analyzed, we did not observe any intrinsic properties of glycopeptides, glycans, or glycosites that appear to associate with an increased sensitivity with one transformation compared to the other. It seems rather that CLR-transformation makes our analyses more sensitive to differences of smaller magnitude. We have added this consideration to the revised figure legend of Supplementary Figure 1. With the currently available public data, and our focus on glycomics data here, it is not feasible to say whether all these glycopeptides that change significance are genuinely true. Yet we hypothesize, given that our CoDA approach does recover genuinely true differences in glycomics data (e.g., Fig. 1b), at least a similar situation will be the case for these glycoproteomics data. But again, we want to emphasize that the applicability of our approach to glycoproteomics data is in no way central to our manuscript and simply presents an additional advantage of our work (in fact, we discuss below why a dedicated glycoproteomics analysis workflow is still needed in the field to properly account for the higher data complexity).

The manuscript introduced the center log-ratio (CLR) and additive log-ratio (ALR) transformations approach to analyze comparative glycomics data. The author analyzed the publicly available glycomics datasets and demonstrated the pitfall of the traditional statistical analyses, which are based on percent abundance normalization as a sum of the whole glycan. CLR and ALR methods proved to be better statistical than the Aitchison simplex method. The glycomics community has been needing a better and more robust quantitative method for glycomics for a long time. The addition of this method for the glycomics analysis will be a major add-on to the subject area.

However, I have a few concerns. One is that CLR-based analysis shows different outcomes from the already published data. Some potential changes in the glycans were nullified, and other

glycans emerged as significant changes. This will eventually redirect the research outcome and future directions.

It is known that data % abundance normalization is biased when introducing artificial changes in abundance value as it is relative to the other part of the data. Thus, it is very important to compare this proposed method and the Aitchison simplex method to a standard sample where known concentration of glycan was endogenously expressed or spiked-in of a different class of the glycans like High-mannose, complex/hybrid, sialylated, and both fucosylated and sialylated type of glycans at different concentration and assess the performance of quantitative analysis.

We agree with this assessment and, as detailed above, have engaged in an investigation of new glycomics data containing data from glycan standards in known absolute concentrations (**new Fig. 1b**). The results of this new analysis have strengthened the confidence that can be placed in result differences that emerge when using our herein developed methods and workflows. We are convinced that this, together with the other analyses and improvements that we have conducted in the course of this revision, has made this new platform for comparative glycomics analysis more transparent and more robust, preparing CoDA-based glycomics analyses to become the new standard in the field.

Reviewer #3 (Remarks to the Author):

Bennett and colleagues demonstrate the application of CLR transformation instead of normalized glycan ratios, which result in compositional data. The authors have made functions available through their Glycowork Python package, which is a highly useful resource for the glycomics field.

The main concern currently is whether the presented work is novel enough for publication in Nature Communications. However, this transformation has been employed in other omics fields before. Furthermore, transformations in glycomics were also discussed and evaluated in this article: Uh, Hae-Won, et al. "Choosing proper normalization is essential for discovery of sparse glycan biomarkers." *Molecular omics* 16.3 (2020).

There is clearly a need for better statistical tools in the glycomics field, as there are likely many false positives and negatives in published data. However, there are many other experimental and technical factors influencing the outcomes of glycomics data. The authors should elaborate on these points.

We thank the reviewer for their evaluation of our work and respond to individual points below. We respectfully disagree with the statement that this work has been evaluated before in the indicated article. Uh et al. have indeed employed ALR as a normalization technique for glycomics data, and we have now added this reference to the revised manuscript, yet: (i) CLR was not assessed, (ii) no formal means of choosing the reference component for ALR was used

(here, we use the Procrustes correlation for this purpose), raising questions about the optimality of the used ALR procedure, (iii) neither scale uncertainty, nor informed-scale variants of CLR/ALR have been used in the indicated work, (iv) subcompositional coherence, guaranteed by CLR/ALR, is crucial for motif analysis and has not been explored by Uh et al. but by our work here, (v) we also assess and apply all these data analysis aspects to a far greater diversity and quantity of glycomics data (including now also to absolute concentrations of glycans in defined mixtures; see **new Fig. 1b**) and applications compared to the indicated work, and (vi) make these new state-of-the-art methods readily available to researchers, via a Python interface and a graphical user interface.

Especially point (iii), the issue of scale uncertainty and informed scale models, is (a) at the heart of our herein presented work, (b) of great practical importance, and (c) entirely novel to the realm of glycomics. We also adapt this framework to the specific use-case of glycomics data (e.g., glycan motif analyses, informed scale models using glycan ion intensities, etc.). In general, transfer of knowledge—for instance in the form of introducing and fine-tuning an established technique into a new field that hitherto used less appropriate analysis methodology—does still exhibit novelty.

We would, however, still argue that novelty is not the only criterion to judge the merit of this work. Importance would be another criterion, as well as applicability. In both those dimensions, we would argue for high scores of our work, since it is applicable to all generated comparative glycomics data, which is an area that is becoming more and more important to, for instance, understand cellular physiology or disease progression.

It is, of course, true that many other factors influence the quality and outcomes of glycomics data and we have revised our discussion to better reflect these considerations.

Minor points

1) -> That's for an abstract difficult to follow, from where does this number come from? "or routine false-positive rates of >25% for differential abundance"

We have revised our abstract to be easier to follow. The specified number comes from our own analyses (e.g., Fig. 1c) and is also supported by similar findings in related 'omics fields that we reference in our work (e.g., Nixon et al., 2024, doi: 10.1101/2024.04.01.587602 and Li et al., 2022, doi: 10.1186/s13059-022-02648-4). We have removed the specific number from the revised abstract.

2) Line 103: "Additionally, as will be shown, the total number of molecules is rarely invariant across conditions, further distorting the obtained relative abundances and their differences across conditions."

-> This is exactly the scenario that would occur when the occupation of a glycosylation site increases from 1% to 100%. The additional glycans will decrease the relative abundance of all

other glycans. Why do the authors not only focus on glycoproteomics data? Isn't it most important to know which protein and which glycosylation site the changes occur at?

We appreciate that glycoproteomics often provides greater functional insights compared to bulk-glycomics. To this end, we have ensured our analysis techniques are suited in principle to glycoproteomics data (e.g., Supplementary Fig. 1). At this point, however, much of the available data (and insights) in the field of glycomics stems from protein-liberated glycans and it is to be expected that this family of methods will continue to be widely used. Additionally, glycomics is not merely a proxy for glycoproteomics; applications such as breast milk oligosaccharides or glycosphingolipids are conceptually not applicable for glycoproteomics but are so for glycomics, ensuring the continued usage of this method. Thus, it is important for the validation of our analyses to reflect the data being generated by researchers in the field and provide analysis methods that are specialized for this type of data.

We further would like to note that, while our differential abundance workflows are sufficient for basic analyses of glycoproteomics data, a much more extensive specialization could—and should—be envisioned for this type of data. The nested nature of glycoprotein-glycosite-glycoforms in glycoproteomics data introduces dependencies and constraints that are not contained in data from released glycans and would require substantial work to properly address. We have added these considerations to the revised discussion.

3) Line 73 and Table 1: “When applied to a range of glycomics datasets, we show that differential expression analysis using this enhanced CLR/ALR transformation can reveal biological insights.”

-> These findings, are difficult to verify. How can we be sure that these are genuine biological insights and not (again) artifacts? This demonstrates the complexity of the field, as small changes in the workflow may provide very different outputs.

This is an important consideration. One argument here is of theoretical basis: Glycomics data are undoubtedly compositional data and CoDA is therefore, from a statistical point of view, more appropriate *a priori* to analyze such data, as the assumptions of the analyses then actually match the data distribution (which is desirable from a fundamental standpoint). We also would like to point here to the fact that CoDA-based approaches allow us to cluster genuine biological groups better than conventional analyses (e.g., Fig. 2a, Fig. 4d), indicating their superior performance.

Of course, in isolation, it is difficult to claim our new findings are more valid than previous reports. We have therefore performed new glycomics experiments, using samples with known absolute-concentrations of glycans, which is included in our revised manuscript as the **new Fig. 1b**, and which shows that our approach remedies the distortions introduced by relative abundances. This clearly demonstrates that our analysis approach objectively outperforms analyzing relative abundances as commonly done today.

4) Reference 6: The referenced article by Gloor and colleagues is a review article, and the methods have been applied earlier in different fields. This should be correctly referenced and represented.

We have now added earlier references of the development and application of CoDA methodologies, including to the work of John Aitchison himself, to better reflect the history of this field.

Reviewer #4 (Remarks to the Author):

The authors present a good solution to a problem that I've run afoul of before without an obvious right way to solve it. This is an important method to share with the community, and to help to draw attention to this recurring issue with statistical analysis.

We thank this reviewer for their assessment of our manuscript and respond to each individual point below. We are confident this has made our revised manuscript clearer and better contextualized.

Major comments

1. There are a lot of places where the mathematical nomenclature uses terms that have a distinct meaning in mass spectrometry-based glycomics analysis. If you want to use the nomenclature that is consistent with CoDA, please make your definitions clear early on.

a. Composition analysis may refer to either the analysis of the number of each of a thing in a population or an analysis of glycans by monosaccharide composition, not structure. Your target audience may not understand which you mean, making it harder for the importance of your methods to come across. It may be that `glycowork` does not have a concept of an unstructured bag of monosaccharides, which might also be clarified.

This is an important point and we have now clarified the distinction between composition as the constituent monosaccharides vs the mathematical term. Glycowork does indeed have a great amount of functionality targeted at pure compositions (e.g., mass calculation, mapping to potential structures, differential abundance analyses, etc.), so this is also relevant in this context.

b. Relative abundance, in the context of a population is the proportion of different components of that population, but in mass spectrometry, relative abundance refers to the ion intensity measured by the instrument, either as a point value in a spectrum or integrated over time. The latter is what most MS-based quantitative experiments use quantify glycans unless isotopically labeled reagents like tandem mass tags (TMT) are used, which is a measure of absolute abundance because it lets us effectively ignore ionization efficiency by quantifying reporter ions from fragmentation instead of the precursor.

We also have distinguished these terms in the revised manuscript to make nomenclature clearer and add more context to our description.

c. The term "feature" is used interchangeably with "glycan", but not all features are glycans, and some features are backed by multiple glycans. Please use explicit language to make it clear which you are referring to.

We have made our terminology here more specific and clearer to always indicate which features are meant.

2. The narrative is fragmented as it tries to touch on at least five datasets with little explanation about what the objective of each study was, which in turn makes understand how or why the metric of success changes. Additionally, after the first comparison shown in Figure 2, the reader is never shown a head-to-head comparison of CoDA to demonstrate how it helps on experimental data with real world noise characteristics.

We have revised the manuscript to explain the goals of the respective studies in more detail and make this part clearer. To some extent a diversity of datasets and different types of studies was precisely the point of our approach, in order to showcase the general nature of our CoDA-based platform, which can be applied to a great range of comparative glycomics cases, including univariate/multivariate, different glycan classes, etc.

We would like to point out, however, that the datasets analyzed in Figure 2 do comprise experimental data with real-world noise characteristics that has been analyzed in a head-to-head comparison. Further, Figure 4d also features a direct comparison between non-CoDA and CoDA approaches. Additionally, all datasets in Table 1 have been compared, with the findings of both conventional analyses and CoDA-based approaches shown. Thus, readers are already shown several instances of relevant head-to-head comparisons. During this revision, we have added another such comparison for the example of the **revised Figure 4c**, again showing the improvements that come with a CoDA-based analysis (as well as the new glycan standards experiment in the **new Fig. 1b**).

3. The scale uncertainty components of ``glycowork.glycan_data.stats.clr_transformation`` and ``alr_transformation`` invoke the global NumPy RNG without any management of the random state or random seed, which means that unless one seeds the RNG immediately prior to invoking a ``glycowork`` function that calls one of them, the result is non-deterministic. How much do the results shown in the paper vary over multiple invocations of these functions?

We agree with the reviewer that this is an important point. First, we would like to mention that, during the revision process, we have now fixed the relevant seeds within glycowork, ensuring reproducibility, which will be part of the upcoming glycowork version 1.4 (already publicly available on the dev branch). Since we were still curious about how much analyses varied in the absence of fixed seeds, we have generated the **new Supplementary Figure 4**, which shows that, at least with the default $\gamma = 0.1$ within glycowork, differential abundance

results remain qualitatively and quantitatively reproducible between analysis runs. However, we would expect that higher values of γ would also result in higher discrepancies between runs.

4. The supplementary tables show only the final tables of differences. No "Source Data" tables showing the data used to produce the plots in the main text, which makes reproducing the analysis even more problematic. It is not obvious if the MissForest algorithm mentioned in the methods is even a component of `glycowork`.

All source data, which have been used as is for the functions in glycowork v1.3, are available from and stored within glycowork v1.3 (available via `glycowork.glycan_data.loader.glycomics_data_loader`; and `.glycoproteomics_data_loader` from v1.4 onward). MissForest is indeed a component of glycowork (`glycowork.glycan_data.stats.MissForest`) and is invoked during preprocessing. All workflows described in this manuscript can be (and have been) entirely executed within glycowork. We have clarified these points in the revised manuscript. In the interest of streamlining and transparency, we have now also centralized all preprocessing for all our analysis functions within one hub function (`glycowork.motif.analysis.preprocess_data`) that will be rolled out in the upcoming glycowork v1.4. Glycowork v1.4 will also include a dedicated tutorial for differential abundance analysis in its updated documentation.

Minor comments

5. The repeated expression of "glycan or glycopeptide" uses a lot of space for not a lot of added value after this is stated for the fourth time. A glycosite is a glycome in microcosm, but it has its own complications. A glycosite spanned by a fully and partially mis-cleaved peptide bearing the same glycan are not truly additive because of differences in peptide sequence ionization efficiency. Also make sure to plug glycolipids if it is suitable for them as well.

We have toned down the number of times the double-application is mentioned and agree with the reviewer. We also now mention the applicability of our methodology to glycolipids in the revised manuscript.

6. The comments regarding ionization efficiency and MS quantification gives the impression that spectrum counting a common form of quantification. I doubt it is reasonable to claim so in isolation.

We have revised and shortened this segment to keep our focus on differential ionization efficiency across glycans to indicate that measured ion intensities are influenced by how well a glycan ionizes. Edge cases of this phenomenon may also manifest as low-abundance glycans slipping below peak detection limit in case of a lower ionization efficiency, or vice versa.

7. The discussion of informed scale models mentions using the ion intensities as a the value of ζ , but it isn't specified whether $\text{mean}(\text{ion}\text{trm}\{\text{glycans}\}_{\text{condition}})$ is the

average of the sum of all glycans in each sample in the condition or some other function. I think the methods section later indicates this is a geometric mean, but that is a source of confusion.

In the revised manuscript, we have clarified that the informed scale models indeed use the average of the summed ion intensity of all glycans in each sample in the condition as an indicator for the total glycan amount in that condition.

Thank you,

Joshua Klein

Reviewer #4 (Remarks on code availability):

We thank the reviewer for helpful feedback on our code quality and are confident that our actions in this regard have improved the overall quality of glycowork as well as the specific code module that comprises this work.

As described in the main commentary, the code as written uses a global RNG that would have to be controlled by the caller to make any function invocation reproducible. Without the program that was used to produce the figures in the paper, along with their final conclusions it would not be possible to determine if a random seed was used.

We have now fixed the global RNG within glycowork, to make outputs deterministic and reproducible. This is already active on the dev branch and will be shortly rolled out in the upcoming glycowork version 1.4.

The code is readable and documented, if difficult to navigate due to the dozens of free functions per file as encouraged by the nbdev programming style. The docstrings are written in a format that is neither the official rST notation, NumPy, nor Google style syntax, instead opting to render them as rST tables, but most functions are documented.

Yes, since we rely on nbdev for our documentation and package organization, we have opted to adhere to the guidelines proposed by nbdev.

The rest of the codebase is functionally solid. Minor quibbles about internal representation types being loosely defined strings and DataFrames aside.

The git repository should not track the top-level `build` or `_proc` directories, and should purge the `__pycache__` directories as well, but this is minor. It may be desirable to replace several of the larger static data files with compressed formats that reduce the overhead when transporting them, e.g. Parquet or compressing the csv files, and to gzip the pickle files.

We thank the reviewer for this suggestion and have updated our .gitignore file as well as purged the repository. These changes are currently active on our public dev branch and will be part of glycowork version 1.4 (expected to come out this fall).

Response to reviewers

Reviewer 1:

The authors have effectively addressed all of our comments and concerns. We appreciate their addition of the constant 2 pmol concentration of standards 1 and 2 (glycan m/z: 893.4 and 941.39) in all samples, while standards 3 and 4 (glycan m/z: 1009.94 and 1111.46) are in increasing order. The pattern of standard glycan concentrations and the skewed nature of relative abundance are well-captured in the ALR-transformed data (Figure 1b). However, we note the negative value for glycan 941.39 in the ALR-transformed data (Figure 1b and Supplementary Table 1). We request an explanation for this negative trend and guidance on how to compensate or adjust for this observation in the analysis to accurately express glycan abundance in real datasets.

Additionally, we suggest clarifying the bar color code in Figure 1b, including the gray color.

We thank the reviewer for their assessment. For the negative values, we would like to add that neither CLR nor ALR are in principle zero-bounded, due to operations such as taking the logarithm. That means that negative values are absolutely expected and, on average, simply translate to “small values”. Specifically with ALR, negative values indicate a value lower than the chosen reference. Here in this case, this also illustrates the measurement issue of differential ionizability we raise in the manuscript, where the neutral 941.39 structure is detected in lower amounts than the negatively charged other structures. We have added an explanation of these negative values in the manuscript. We have also made the color in Fig. 1b clearer.

Reviewer 4:

A dedicated RNG is now created and seeded on module import, but it is still not controlled at the call site of ``clr_transform`` and friends, and worse, it is now even harder for the caller to control! As the authors addressed my concerns about the stability of the method w.r.t. the random element in the revision, the following is a suggestion for code quality, not a requirement for revision.

To control it, the user must explicitly write ``glycoworks.glycan_data.stats.rng = np.random.default_rng(...)``. Instead, the caller should be able to pass a ``RandomState``/RNG-seed-like value as an optional parameter to produce an RNG for the function and all its callees to use, falling back to some default seed/shared RNG state when the argument is absent. This is a pervasive pattern in the ``scikit-learn`` library.

This does involve piping a new parameter throughout the call-graph which the authors may find too odious to write, in which case I suggest adding a note to the documentation about the shared RNG, and adding a top-level function that can update the shared RNG without breaking encapsulation.

This is a good point and we have indeed now added a “`random_state``” keyword argument to the CLR-transformation to allow users to provide a reproducible RNG state. This is already available on the public developer’s version of glycowork and will be rolled out with the v1.5 update (Spring 2025).